# DUAL ALIGNMENT FOR COVARIATE SHIFT: A PRINCIPLED FRAMEWORK FOR GRAPH DOMAIN ADAPTATION

## ABSTRACT

*Graph Domain Adaptation* (*GDA*) is fundamentally challenged by *Covariate Shift* (*CS*), a pervasive discrepancy between source and target graph distributions. We decompose CS into two complementary components: *Feature Shift* (*FS*), arising from mismatched node feature distributions, and *Feature-Conditional Structure Shift* (*FCSS*), reflecting structural variations conditioned on features. Both FS and FCSS distort *Graph Neural Network* (*GNN*) representations, thereby hindering reliable cross-domain transfer. To overcome these issues, we propose *Dual Alignment for Covariate Shift* (*DACS*), a framework that jointly addresses FS and FCSS through adversarial feature alignment for domain-invariant embeddings and adaptive reweighting to enforce structural consistency. Extensive experiments on benchmark datasets demonstrate that DACS effectively bridges domain gaps and consistently outperforms state-of-the-art baselines, highlighting its strong cross-domain generalization.

## 1 INTRODUCTION

Distribution shifts are inevitable in real-world graph data, making domain adaptation a fundamental challenge for reliable graph learning. *Graph Domain Adaptation* (*GDA*), also known as *cross-network classification* (Shen et al., 2020; Shi et al., 2023b), aims to transfer knowledge from a labeled source graph domain to an unlabeled or sparsely labeled target domain. This enables generalizable node classification even in the presence of domain shifts. GDA is widely applicable in settings where graph-structured data varies across domains, such as social network analysis (Liu et al., 2021; Xiao et al., 2025), protein interaction prediction (Li et al., 2024), and traffic forecasting (Xu et al., 2024; Shao et al., 2025). In practice, such applications often involve shifts in node features, graph structures, or both.

To systematically characterize these shifts, (Liu et al., 2024b) introduced the notion of *Covariate Shift* (*CS*), which formalizes discrepancies in the joint distribution of node features and graph structures. A representative case is cross-platform social network analysis: user interaction patterns (graph structure) and interest profiles (node features) differ greatly between platforms such as Twitter and LinkedIn, but the relationship between these attributes and engagement behaviors (labels) is relatively stable. Thus, CS provides a natural and general lens for studying domain shifts in GDA.

While CS adaptation has been widely studied in non-graph settings (Shimodaira, 2000; Segovia-Martín et al., 2023), addressing CS in GDA is more challenging because feature and structure shifts often occur simultaneously. These dual shifts reinforce one another: errors caused by mismatched features propagate through message passing, while structural discrepancies further distort learned representations. Empirical findings (Liu et al., 2024b) show that neglecting either shift leads to severe performance degradation—ignoring feature shifts results in large accuracy drops, while ignoring structural shifts undermines generalization. These observations highlight the need for CS-aware methods that disentangle and address both sources of shift.

Early GDA approaches leverage *Graph Neural Networks* (*GNNs*) for cross-domain representation learning, and can be broadly grouped into two categories. *(i)* Representation alignment methods minimize explicit discrepancy metrics, such as maximum mean discrepancy (Shen et al., 2021; Shi et al., 2023a), optimal transport distance (Zhao et al., 2024b), subtree discrepancy (Wu et al., 2023),

or mean squared error between attribute- and topology-derived embeddings (Fang et al., 2025a). *(ii)* Adversarial learning methods encourage encoders to produce domain-invariant embeddings by training against domain discriminators (Zhang et al., 2019; Wu et al., 2020; Dai et al., 2023). More recent work incorporates regularization strategies, such as spectral regularization to bound target risk (You et al., 2023), or selective propagation layer removal to enhance transferability (Liu et al., 2024a). Despite their progress, these methods generally lack explicit mechanisms for handling structural discrepancies, often entangling feature and structure adaptation in ways that yield suboptimal performance.

To address structure-specific shifts, recent studies (Liu et al., 2023; 2024c) proposed the notion of *label-conditional structure shift*, which characterizes cross-domain differences in how node labels influence connectivity. Their solution applies edge reweighting based on ratios of label-conditional edge probabilities. While effective in some cases, this strategy faces two key limitations. First, it relies on pseudo-labels for target-domain nodes, introducing additional noise from label prediction errors. Second, the reweighting factors depend solely on labels and remain static across layers, overlooking the dynamic evolution of node representations through GNN transformations.

In this work, we tackle the challenge of GDA through the lens of *Covariate Shift* (*CS*). We formally decompose CS into two complementary components: *Feature Shift* (*FS*), which captures mismatches in node feature distributions, and *Feature-Conditional Structure Shift* (*FCSS*), which captures structural discrepancies conditioned on features. Our theoretical analysis demonstrates that FS disrupts the consistency of initial representations, while FCSS distorts conditional GNN embeddings, jointly leading to severe representation misalignment.

To address these challenges, we introduce *DACS* (*Dual Alignment for Covariate Shift*), a principled framework that jointly mitigates FS and FCSS through three key modules: *(i)* adversarial feature alignment to eliminate FS and learn domain-invariant embeddings; *(ii)* adaptive layer-wise reweighting to counter FCSS by aligning conditional expectations across domains; and *(iii)* final-layer representation alignment to unify feature and structure adaptation, thereby achieving comprehensive covariate shift reduction.

Our contributions are threefold: *(i)* We establish a principled decomposition of CS into FS and FCSS, and provide a theoretical analysis of their distinct effects on GDA risk. *(ii)* We propose DACS, a dual-alignment algorithm that integrates adversarial feature alignment, adaptive reweighting, and final-layer alignment to jointly handle FS and FCSS. *(iii)* We conduct extensive experiments on benchmark datasets, demonstrating that DACS substantially outperforms state-of-the-art baselines, with ablation studies further validating the complementary roles of FS and FCSS mitigation.

## 2 Preliminaries

### 2.1 Node Classification

Let $\mathcal{G} = (\mathcal{V}, \mathcal{E})$ denote an undirected graph, where $\mathcal{V}$ is the set of nodes with $|\mathcal{V}| = n$ and $\mathcal{E}$ is the set of edges. The adjacency matrix is denoted by $\boldsymbol{A} \in \mathbb{R}^{n \times n}$. Each node $v_i \in \mathcal{V}$ is associated with a feature vector $\boldsymbol{x}_i \in \mathbb{R}^F$, and all node features together form a matrix $\boldsymbol{X} \in \mathbb{R}^{n \times F}$, where $F$ is the input feature dimension.

We study the node-level classification task (also known as cross-network classification), which aims to predict the label vector $\boldsymbol{Y} = (Y_u)_{u \in \mathcal{V}}$. For theoretical clarity, we assume the binary case with $Y_u \in \{0, 1\}$, though our framework naturally extends to multi-class settings. The joint distribution of features, structure, and labels is denoted by $P(\boldsymbol{X}, \boldsymbol{A}, \boldsymbol{Y})$, with marginal distribution $P(\boldsymbol{X}, \boldsymbol{A})$ and conditional label distribution $P(\boldsymbol{Y} \mid \boldsymbol{X}, \boldsymbol{A})$. We define the ground-truth labeling function as $f_P : \mathbb{R}^{n \times F} \times \mathcal{A} \to \{0, 1\}^n$, which assigns labels to all nodes. The classification *risk* of a predictor $f$ under distribution $P$ is

$$\mathcal{R}_P(f) := \mathbb{E}_P \big\| f(\boldsymbol{X}, \boldsymbol{A}) - f_P(\boldsymbol{X}, \boldsymbol{A}) \big\|_1 = \mathbb{E}_P \sum_{u \in \mathcal{V}} \big| f_u(\boldsymbol{X}, \boldsymbol{A}) - f_{P,u}(\boldsymbol{X}, \boldsymbol{A}) \big|,$$

where $f_u(\boldsymbol{X}, \boldsymbol{A})$ and $f_{P,u}(\boldsymbol{X}, \boldsymbol{A})$ are the predicted and true labels for node $u$, respectively.

*Graph Neural Networks* (*GNNs*) (Kipf & Welling, 2017; Wu et al., 2019) have become the standard architecture for graph representation learning. A GNN updates node representations through iterative neighbor aggregation and feature transformation. The layer-wise propagation rule is

$H^{l+1} := \phi^\ell (A H^\ell)$, where $H^\ell$ denotes node features at layer $\ell$, $\phi^\ell$ denotes the $\ell$-th layer encoder. With $H^0 = X$, the final embedding after $(L-1)$ layers is $H^L = \phi(X, A)$. A classifier $g$ then predicts labels from these embeddings, i.e., $Y = g(H^L)$. Hence, the overall classification model can be expressed as the composition $g \circ \phi$.

## 2.2 COVARIATE SHIFT IN GRAPH DATA

Existing studies (Liu et al., 2023; 2024c) primarily attribute structural shift to domain-specific label–structure relations, yet the role of *feature-driven* structural variation remains insufficiently explored. In many real-world graphs, however, node features often exert a more fundamental influence on edge connectivity than node labels. For instance: *(i)* in sociological and biological networks, nodal attributes are the primary determinants of structural patterns (Fosdick & Hoff, 2015); *(ii)* in brain networks, fMRI signals of different regions better explain functional connectivity than diagnostic labels (Fang et al., 2025b); and *(iii)* in social networks, user attributes such as age and interests largely drive link formation beyond categorical group labels (Kim & Leskovec, 2011). These observations indicate that structural discrepancies across domains may arise not only from label-conditioned mechanisms, but also from differences in how features shape connectivity.

To formalize this broader view, we adopt the notion of *Covariate Shift (CS)* in graph data (Liu et al., 2024b).

**Definition 2.1** (*Covariate Shift*). Covariate shift occurs when the joint distribution of node features $X$ and adjacency matrix $A$ differs between the source and target domains: $P_S(X, A) \neq P_T(X, A)$.

This definition naturally encompasses shifts in both node features and graph structures, phenomena frequently observed in citation networks, social networks, and biological systems. From Definition 2.1, two direct forms of shift can be identified: *(i) Feature Shift* (*FS*): mismatches in marginal feature distributions, $P_S(X) \neq P_T(X)$; *(ii) Structure Shift* (*SS*): mismatches in marginal structural distributions, $P_S(A) \neq P_T(A)$.

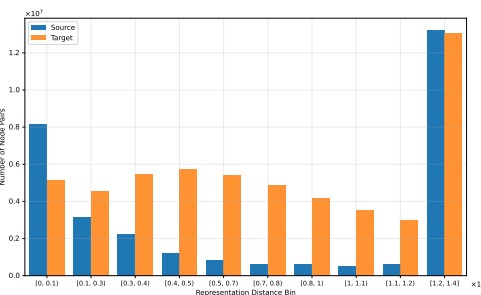 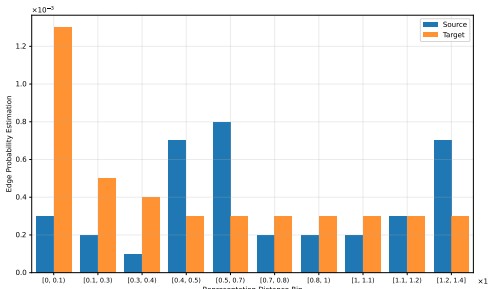

(a) FS: discrepancies in node feature distributions across domains.

(b) FCSS: domain-specific edge probabilities given similar representations.

Figure 1: Visualization of CS components between `DBLPv8` and `ACMv9`.

However, CS may persist even when both FS and SS are absent. By factorizing the joint distribution as $P(X, A) = P(X) P(A \mid X)$, we uncover the possibility of *Feature-Conditional Structure Shift* (*FCSS*): discrepancies in how node features govern connectivity across domains, i.e., $P_S(A \mid X) \neq P_T(A \mid X)$. This implies that CS can occur even when feature marginals and structural marginals are perfectly aligned, as long as the feature–structure dependency differs.

Taken together, covariate shift in graphs can thus be more fundamentally understood through two complementary components: FS and FCSS. Figure 1(a) illustrates FS by showing the number of node pairs whose feature distances fall into different intervals in both domains, while Figure 1(b) illustrates FCSS by comparing, for each feature-distance interval, the proportion of connected node pairs between real-world datasets DBLP and ACM. To further illustrate FCSS, in the left subfigure of Figure 2, we visualise a five-node synthetic example and explicitly compute the conditional edge probabilities associated with each pairwise feature distance in both domains, exhibiting the domain-variant feature-structure dependence.

Compared with the conventional FS–SS decomposition, this perspective offers a deeper characterization of distribution shifts in graph data, as FCSS explicitly captures structural variations conditioned on node features.

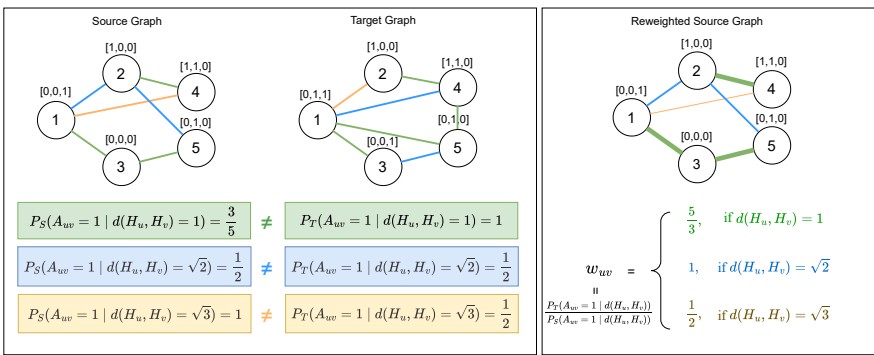

Figure 2: The left subfigure shows a five-node synthetic graph, where Nodes 1 and 3 change their features across domains, illustrating FS. In the source graph, five node pairs (1-3, 2-3, 2-4, 3-5, 4-5) have a feature distance of 1, among which three are connected by edges, yielding a conditional edge probability of $3/5$. In the target domain, the conditional edge probability for distance 1 is $4/4 = 1$. This discrepancy in conditional edge probabilities reflects FCSS. The right subfigure calculates the edge probability ratio across domains, which is used as a weight multiplied to the source edge weights. From the thicker or thinner edges, we observe that we observe this reweighting reinforces edges that are more likely to appear in the target graph and attenuates those that are less likely.

# 3 GRAPH DOMAIN ADAPTATION UNDER COVARIATE SHIFT

*Graph Domain Adaptation* (*GDA*) considers a learning scenario where a GNN is trained on a source domain but deployed on a target domain with different data distributions. Formally, we denote a labeled source-domain graph as $(\boldsymbol{X}, \boldsymbol{A}, \boldsymbol{Y}) \sim P_S$, where $\boldsymbol{X}$ represents node features, $\boldsymbol{A}$ the adjacency matrix, and $\boldsymbol{Y}$ the node labels. The target domain provides only unlabeled graphs $(\boldsymbol{X}, \boldsymbol{A}) \sim P_T$, drawn from a distribution $P_T$ that differs from $P_S$ in its statistical properties. For simplicity, we denote the labeling functions of the two domains as $f_S := f_{P_S}$ and $f_T := f_{P_T}$.

To facilitate adaptation under distributional shifts, we introduce a domain-shared multi-layer perceptron (MLP) $\phi^0$ that preprocesses node features before GNN aggregation, producing $\boldsymbol{H}^1 = \phi^0(\boldsymbol{X})$ as the input to the GNN. This generalizes the conventional design where raw features are directly used (i.e., $\boldsymbol{H}^1 = \boldsymbol{X}$ in Section 2.1) and provides additional flexibility to align representations across domains. The goal of GDA is to learn a GNN classifier $g \circ \phi$ that minimizes the target risk $\mathcal{R}_T(\cdot)$ under CS. If no shift exists (i.e., $P_S = P_T$), training solely on the source domain would suffice. However, when CS is present, it is crucial to understand how it contributes to the target risk. To this end, we derive an upper bound on the target risk $\mathcal{R}_T(g \circ \phi)$, which reveals how FS and FCSS influence adaptation.

**Theorem 3.1.** *Under the CS setting in Definition 2.1, consider the classifier $g \circ \phi$ with initial representation $\boldsymbol{H}^1 = \phi^0(\boldsymbol{X})$ and final representation $\boldsymbol{H}^L = \phi(\boldsymbol{H}^1, \boldsymbol{A})$ as defined in Section 2.1. Then, the classification risk on the target domain satisfies*

$$\mathcal{R}_T(g \circ \phi) \leq \underbrace{\mathcal{R}_S(g \circ \phi)}_{\textit{Term (I)}} + \underbrace{\int_{\mathcal{H}^1} r_S(\boldsymbol{H}^1) \cdot \left| d\,\mathrm{P}_T(\boldsymbol{H}^1) - d\,\mathrm{P}_S(\boldsymbol{H}^1) \right|}_{\textit{Term (II)}}$$

$$+ \underbrace{\mathbb{E}_{P_T}\left( \int_{\mathcal{H}^L} r_S(\boldsymbol{H}^1, \boldsymbol{H}^L) \cdot \left| d\,\mathrm{P}_T(\boldsymbol{H}^L \,|\, \boldsymbol{H}^1) - d\,\mathrm{P}_S(\boldsymbol{H}^L \,|\, \boldsymbol{H}^1) \right| \right)}_{\textit{Term (III)}}. \quad (1)$$

*Here, $r_S(\boldsymbol{H}^1, \boldsymbol{H}^L) := \mathbb{E}_{P_S}\big( \|g(\boldsymbol{H}^L) - f_S(\boldsymbol{X}, \boldsymbol{A})\|_1 \,|\, \boldsymbol{H}^1, \boldsymbol{H}^L \big)$ is the risk when the initial and final representations are $\boldsymbol{H}^1$ and $\boldsymbol{H}^L$ in the source domain, and $r_S(\boldsymbol{H}^1) := \mathbb{E}_{P_S}\big( \|g(\boldsymbol{H}^L) - f_S(\boldsymbol{X}, \boldsymbol{A})\|_1 \,|\, \boldsymbol{H}^1 \big)$ is the risk when the initial representation is $\boldsymbol{H}^1$ in the source domain.*

The target risk bound in Eq. (1) consists of three terms, each reflecting a different factor in domain adaptation:

***(I) Source domain classification risk.*** This term measures the inherent performance of the classifier on the source domain. It reflects the base accuracy achievable without distributional shift and thus sets a lower bound for transfer performance.

***(II) Initial representation discrepancy.*** This term captures the divergence between $P_S(\boldsymbol{H}^1)$ and $P_T(\boldsymbol{H}^1)$, which originates from FS. Specifically, if FS is absent (i.e., $P_S(\boldsymbol{X}) = P_T(\boldsymbol{X})$), then the initial representation distributions align, $P_S(\boldsymbol{H}^1) = P_T(\boldsymbol{H}^1)$. Thus, FS is the direct cause of discrepancies in the initial representations. To reduce this divergence, we employ adversarial feature alignment (Ganin et al., 2016) to train the encoder $\phi^0$, as detailed in Section 5.1.

***(III) Final representation discrepancy.*** This term measures the gap between $P_S(\boldsymbol{H}^L \mid \boldsymbol{H}^1)$ and $P_T(\boldsymbol{H}^L \mid \boldsymbol{H}^1)$, which arises fundamentally from FCSS. In the absence of FCSS, the conditional adjacency distributions align, $P_S(\boldsymbol{A} \mid \boldsymbol{X}) = P_T(\boldsymbol{A} \mid \boldsymbol{X})$, which guarantees that $P_S(\boldsymbol{H}^L \mid \boldsymbol{H}^1) = P_T(\boldsymbol{H}^L \mid \boldsymbol{H}^1)$ (see Proposition A.1 in Appendix A). Therefore, FCSS is the necessary condition for final representation divergence. To mitigate this effect, we introduce a layer-wise reweighting strategy on source representations (Sections 4 & 5.2) and further align the reweighted source and target representations at the final layer (Section 5.3), ensuring consistency in the conditional distributions across domains.

## 4    REWEIGHTING FOR CONDITIONAL EXPECTATION ALIGNMENT

To address the final-layer representation discrepancy highlighted by the last term in Eq. (1), we propose a *layer-wise reweighting framework* designed to suppress the effect of FCSS. The central idea is to adaptively reweight the source-domain message-passing process such that the conditional expectations of aggregated representations become consistent with those of the target domain.

Concretely, we introduce a *conditional probability ratio* as the reweighting factor $w^\ell > 0$ at each GNN layer $\phi^\ell$. Starting from $\boldsymbol{H}_w^1 := \boldsymbol{H}^1$, the reweighted propagation is defined recursively as $\boldsymbol{H}_w^{\ell+1} := \phi^\ell\big(w^\ell \boldsymbol{A} \boldsymbol{H}_w^\ell\big)$, for all $\ell \in [L-1]$. Here, the weight factor is given by

$$w^\ell := \frac{P_T(\boldsymbol{A} \mid \boldsymbol{H}_w^\ell, \boldsymbol{H}^1)}{P_S(\boldsymbol{A} \mid \boldsymbol{H}_w^\ell, \boldsymbol{H}^1)},$$

where $P_T$ and $P_S$ denote the edge probabilities conditioned on $(\boldsymbol{H}_w^\ell, \boldsymbol{H}^1)$ under the target and source domains, respectively. Intuitively, this ratio corrects for domain-specific biases in feature-conditioned connectivity, ensuring that the source aggregation process reflects the same conditional structure as the target.

The following theorem establishes that such reweighting preserves conditional expectation alignment across layers.

**Theorem 4.1.** *If the representation at layer $\ell$ has aligned conditional expectations across domains, i.e., $\mathbb{E}_{P_T}(\boldsymbol{H}^\ell \mid \boldsymbol{H}^1) = \mathbb{E}_{P_S}(\boldsymbol{H}_w^\ell \mid \boldsymbol{H}^1)$, then the aggregated representations also align:*

$$\mathbb{E}_{P_T}(\boldsymbol{A} \boldsymbol{H}^\ell \mid \boldsymbol{H}^1) = \mathbb{E}_{P_S}(w^\ell \boldsymbol{A} \boldsymbol{H}_w^\ell \mid \boldsymbol{H}^1).$$

Theorem 4.1 provides the theoretical foundation for our reweighting strategy. If the output of $\phi^\ell$ preserves expectation alignment given mean-invariant inputs, then alignment at layer $\ell$ guarantees alignment at layer $\ell + 1$. By induction, this yields domain-invariant conditional expectations at the final layer: $\mathbb{E}_{P_T}(\boldsymbol{H}^L \mid \boldsymbol{H}^1) = \mathbb{E}_{P_S}(\boldsymbol{H}_w^L \mid \boldsymbol{H}^1)$. Thus, the proposed reweighting framework systematically eliminates the message-passing discrepancy caused by FCSS, leading to aligned high-level representations across domains. In particular, it directly targets the third term in the target risk bound Eq. (1), ensuring that the final representation divergence induced by FCSS is effectively suppressed.

## 5    DUAL-ALIGNMENT ALGORITHM FOR GRAPH DOMAIN ADAPTATION

We propose *DACS (dual-alignment for covariate shift)*, a unified framework that jointly aligns FS and FCSS in graph domain adaptation. Guided by the theoretical insights of Section 3.1, the algorithm integrates three complementary modules: *(i) Feature alignment* via adversarial training

to mitigate FS (Section 5.1); *(ii) Representation reweighting* through adaptive edge-wise corrections to suppress FCSS (Section 5.2); and *(iii) Conditional alignment* of reweighted source and target representations at the final layer to ensure distributional consistency while preserving label discriminability in the source domain (Section 5.3). Together, these three components form a coordinated strategy: adversarial alignment eliminates marginal feature discrepancies, reweighting dynamically adjusts message passing to counter feature-driven structural variations, and final-layer conditional alignment enforces full distributional invariance. This tripartite design enables effective cross-domain generalization under CS. The overall pipeline is presented in Figure 3.

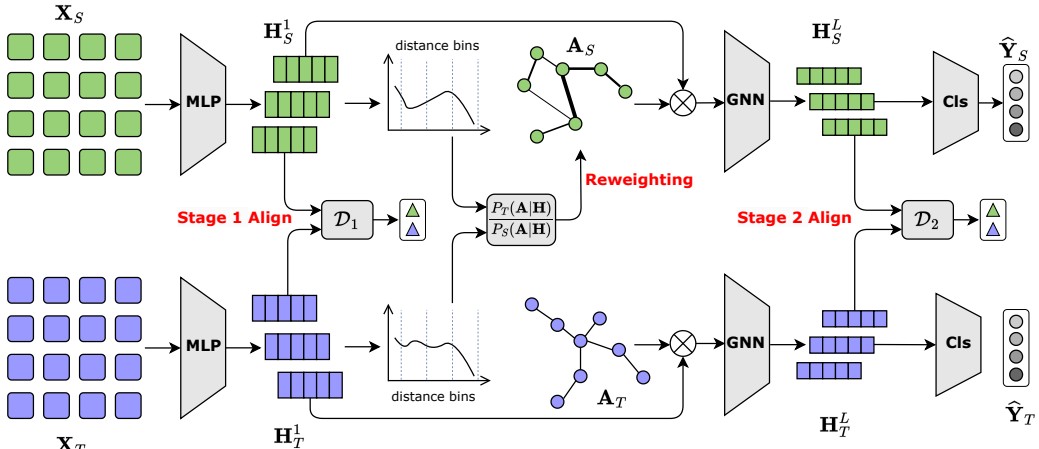

Figure 3: Our dual-alignment GDA pipeline. $\mathcal{D}_1$ and $\mathcal{D}_2$ denote domain discriminators.

## 5.1 ADDRESSING FS VIA ADVERSARIAL TRAINING

FS, corresponding to Term *(II)* in the target risk bound Eq. (1), arises from mismatches in the marginal feature distributions $P_S(\boldsymbol{X})$ and $P_T(\boldsymbol{X})$. To mitigate this, we adopt the *Domain Adversarial Neural Network* (*DANN*) framework (Ganin et al., 2016).

The objective is to learn an encoder $\phi^0 : \mathbb{R}^F \to \mathbb{R}^d$ that maps raw features into a domain-invariant representation space, where $d$ is the embedding dimension. This is achieved through adversarial interaction with a discriminator $\psi : \mathbb{R}^d \to [0,1]$ trained to distinguish between source and target features. The encoder $\phi^0$ is optimized to fool the discriminator, thereby enforcing feature-level alignment. Formally, the adversarial training is expressed as the following min–max optimization problem:

$$\min_{\phi^0}\left[\max_{\psi}\left(\frac{1}{|\mathcal{V}_S|}\sum_{u\in\mathcal{V}_S}\log\psi\big(\phi^0(\boldsymbol{X}_u)\big) + \frac{1}{|\mathcal{V}_T|}\sum_{u\in\mathcal{V}_T}\log\big(1-\psi(\phi^0(\boldsymbol{X}_u))\big)\right)\right], \qquad (2)$$

denoted as $\min_{\phi^0}\mathcal{R}_{\text{FA}}(\phi^0)$. By minimizing $\mathcal{R}_{\text{FA}}$, the encoder learns to produce aligned initial representations $\boldsymbol{H}^1 = \phi^0(\boldsymbol{X})$ across domains. This reduces the representation gap caused by FS and improves the generalization ability of the subsequent GNN layers on the target domain.

## 5.2 MITIGATING FCSS VIA REPRESENTATION REWEIGHTING

Theorem 4.1 shows that reweighting the source-domain GNN representations $\boldsymbol{H}_w^L$ using appropriately defined factors $w^\ell$ aligns their conditional expectations with those of the target-domain representations $\boldsymbol{H}^L$. To operationalize this result, we design a fine-grained, layer-wise adaptive reweighting strategy that corrects for FCSS. Specifically, for each node $u \in \mathcal{V}_S$ and GNN layer $\ell$, we construct an adaptive weight matrix $\boldsymbol{w}^\ell = (w_{uv}^\ell)_{u,v\in\mathcal{V}}$, where each entry is defined as a conditional probability ratio:

$$w_{uv}^\ell := \frac{P_T(A_{uv} \mid \boldsymbol{H}_w^\ell, \boldsymbol{H}^1)}{P_S(A_{uv} \mid \boldsymbol{H}_w^\ell, \boldsymbol{H}^1)}.$$

The reweighted representations are then updated via $\boldsymbol{H}_{\boldsymbol{w}}^{\ell+1} := \phi^\ell((\boldsymbol{w}^\ell \odot \boldsymbol{A})\boldsymbol{H}_w^\ell)$, where $\odot$ denotes the Hadamard product.

To make estimation tractable, we approximate the likelihood term $P_T(A_{uv} \mid \boldsymbol{H}_w^\ell, \boldsymbol{H}^1)$ with $P_T(A_{uv} \mid \boldsymbol{H}_{w,u}^\ell, \boldsymbol{H}_{w,v}^\ell)$. This approximation is well-justified because the probability of an edge is primarily determined by the latent features of its two local endpoints rather than by global graph-level information—a modeling principle widely established in statistical network models such as stochastic block models (Holland et al., 1983), exponential random graph models (Snijders et al., 2006), and latent feature models (Miller et al., 2009). Then, we further model edge likelihoods as a function of pairwise distances $d(\boldsymbol{H}_{w,u}^\ell, \boldsymbol{H}_{w,v}^\ell)$. This leads to the approximation

$$w_{uv}^\ell \approx \frac{P_T(A_{uv} \mid d(\boldsymbol{H}_{w,u}^\ell, \boldsymbol{H}_{w,v}^\ell))}{P_S(A_{uv} \mid d(\boldsymbol{H}_{w,u}^\ell, \boldsymbol{H}_{w,v}^\ell))}.$$

The main justification for this approximation is that complex high-order and non-metric structural patterns affecting edge formation are already absorbed into the GNN embeddings during training, making them geometrically representable through distances or similarities in the embedding space. We estimate $P_T$ and $P_S$ through a distance binning scheme. Concretely, all pairwise distances $\{d(\boldsymbol{H}_{w,u}^\ell, \boldsymbol{H}_{w,v}^\ell) : (u,v) \in \mathcal{V}^2, u \neq v\}$ are partitioned into $J$ disjoint intervals $\{[b_j, b_{j+1})\}_{j=1}^J$. For any $(u,v) \in \mathcal{V}_S^2$, the empirical estimator $\widehat{P}_T(A_{uv} \mid d(\boldsymbol{H}_{w,u}^\ell, \boldsymbol{H}_{w,v}^\ell))$ is defined by

$$\sum_{j=1}^J \mathbf{1}\{d(\boldsymbol{H}_{w,u}^\ell, \boldsymbol{H}_{w,v}^\ell) \in [b_j, b_{j+1})\} \cdot \frac{\sum_{u' \neq v'} \mathbf{1}\{A_{u'v'} = A_{uv}, d(\boldsymbol{H}_{w,u'}^\ell, \boldsymbol{H}_{w,v'}^\ell) \in [b_j, b_{j+1})\}}{\sum_{u' \neq v'} \mathbf{1}\{d(\boldsymbol{H}_{w,u'}^\ell, \boldsymbol{H}_{w,v'}^\ell) \in [b_j, b_{j+1})\}}.$$

For efficiency on large graphs, we can also use a subsampled set of node pairs to estimate probabilities $\widehat{P}_T$. An analogous construction yields $\widehat{P}_S$. The estimated weight $\widehat{w}_{uv}^\ell$ is then defined as a convex combination of the ratio and 1, controlled by a hyperparameter $\lambda \in (0,1)$:

$$\widehat{w}_{uv}^\ell := \lambda \cdot \prod_{v \in \mathcal{V}} \frac{\widehat{P}_T(A_{uv} \mid d(\boldsymbol{H}_{w,u}^\ell, \boldsymbol{H}_{w,v}^\ell))}{\widehat{P}_S(A_{uv} \mid d(\boldsymbol{H}_{w,u}^\ell, \boldsymbol{H}_{w,v}^\ell))} + (1 - \lambda).$$

Finally, the estimated weights are integrated into the GNN propagation across layers: $\boldsymbol{H}_{\widehat{\boldsymbol{w}}}^{\ell+1} := \phi^\ell((\widehat{\boldsymbol{w}}^\ell \odot \boldsymbol{A})\boldsymbol{H}_w^\ell)$. To provide an intuitive illustration of which edge patterns are upweighted or downweighted, we visualise a five-node graph, the calculation of the conditional edge probabilities, and the reweighted source graph with the true probability ratio $w_{uv}$ in Figure 2. The visualization shows that source-domain node pairs likely to be connected in the target domain receive increased edge weights, while those less likely to connect receive decreased weights.

This progressive, edge-specific reweighting directly counteracts FCSS by adapting the aggregation process at each layer. Unlike prior approaches (Liu et al., 2023; 2024c) that rely on pseudo-labels and static weights shared across layers, our method provides dynamic and context-aware adjustments, enabling more expressive message passing and better adaptation to domain-specific structural variations.

### 5.3 Conditional Alignment of Reweighted GNN Representations

In Section 5.2, we showed how reweighting aligns the *first-moment statistics* of source and target representations by matching their conditional expectations: $\mathbb{E}_{P_S}(\boldsymbol{H}_{\boldsymbol{w}}^L \mid \boldsymbol{H}^1) = \mathbb{E}_{P_T}(\boldsymbol{H}^L \mid \boldsymbol{H}^1)$. While this ensures mean-level consistency, it does not capture higher-order discrepancies. To achieve full distributional invariance, we aim for a stricter alignment condition: $P_S(\boldsymbol{H}_{\boldsymbol{w}}^L \mid \boldsymbol{H}^1) = P_T(\boldsymbol{H}^L \mid \boldsymbol{H}^1)$. Together with the feature alignment module in Section 5.1, which guarantees $P_S(\boldsymbol{H}^1) = P_T(\boldsymbol{H}^1)$, the law of total probability implies that the final-layer marginals also align: $P_S(\boldsymbol{H}_{\boldsymbol{w}}^L) = P_T(\boldsymbol{H}^L)$.

To instantiate this principle, we adopt an adversarial training strategy that refines both the feature encoder $\phi^0$ and the GNN layers $\{\phi^\ell\}_{\ell=1}^{L-1}$. A discriminator $\xi : \mathcal{H}^L \to [0,1]$ is trained to distinguish between reweighted source and target representations, while the encoder is optimized to fool $\xi$. The objective is

$$\min_{\{\phi^\ell\}} \left[ \max_\xi \left( \frac{1}{|\mathcal{V}_S|} \sum_{u \in \mathcal{V}_S} \log \xi(\boldsymbol{H}_{\widehat{\boldsymbol{w}},u}^L) + \frac{1}{|\mathcal{V}_T|} \sum_{u \in \mathcal{V}_T} \log\left(1 - \xi(\boldsymbol{H}_u^L)\right) \right) \right], \tag{3}$$

denoted as $\min_{\{\phi^\ell\}} \mathcal{R}_{\mathrm{CA}}(\{\phi^\ell\}_{\ell=0}^{L-1})$. This enforces conditional distribution alignment at the final layer.

In parallel, we retain source-domain supervision to preserve discriminability. The classifier $g$ is trained using cross-entropy (CE) loss:

$$\mathcal{R}_{\mathrm{CE}}(\{\phi^\ell\}_{\ell=0}^{L-1}, g) = -\frac{1}{|\mathcal{V}_S|} \sum_{u \in \mathcal{V}_S} \mathcal{L}_{\mathrm{CE}}\big(g_u(\boldsymbol{H}_{\hat{\boldsymbol{w}}}^L), Y_u\big), \tag{4}$$

where $g_u(\boldsymbol{H}_{\hat{\boldsymbol{w}}}^L)$ denotes the predicted label distribution for node $u$.

The overall training objective integrates feature alignment, conditional alignment, and supervised learning:

$$\min_{\{\phi^\ell\}, g} \Big[ \underbrace{\mathcal{R}_{\mathrm{CE}}(\{\phi^\ell\}, g)}_{\text{Supervised loss}} + \underbrace{\gamma_{\mathrm{FA}} \mathcal{R}_{\mathrm{FA}}(\phi^0)}_{\text{Feature alignment}} + \underbrace{\gamma_{\mathrm{CA}} \mathcal{R}_{\mathrm{CA}}(\{\phi^\ell\})}_{\text{Conditional alignment}} \Big], \tag{5}$$

with $\gamma_{\mathrm{FA}}, \gamma_{\mathrm{CA}} > 0$ as balancing hyperparameters. This joint optimization produces the encoder $\{\phi^\ell\}_{\ell=0}^{L-1}$ and classifier $g$. At inference, predictions on the target domain are made via $g \circ \phi(\phi^0(\boldsymbol{X}), \boldsymbol{A})$. The full algorithmic procedure is summarized in Algorithm 1 and detailed in Appendix B.

## 6 EXPERIMENTS

**Baselines.** We compare our DACS method with the following two categories of methods. *(i)* GDA methods aggregating node features through graph structures: UDA-GCN (Wu et al., 2020), ASN (Zhang et al., 2021), AdaGCN (Dai et al., 2023), CWGCN (Wang et al., 2023), JHGDA (Shi et al., 2023a), and GAA (Fang et al., 2025a). They combine traditional DA approaches with specialized GNN encoders to learn transferable node representations. *(ii)* GDA methods with tailored structure shift mitigation: StruRW (Liu et al., 2023) and PairAlign (Liu et al., 2024c). Implementation details can be found in Appendix C.

**Synthetic Experiments.** To evaluate the performance of our DACS method under different scenarios, we construct a synthetic graph dataset that exhibits three types of covariate shift. In the source domain, node features are drawn from three Gaussian components $P_0 = \mathcal{N}([-1, 0], I)$, $P_1 = \mathcal{N}([1, 0], I)$, and $P_2 = \mathcal{N}([3, 2], I)$. We run $k$-means with three clusters on all features and use the cluster assignments as node labels, so labels are entirely determined by the geometry of $\boldsymbol{X}$. Given these labels, we generate edges with a stochastic block model (SBM) (Abbe, 2018): two nodes are connected with probability $p$ if they share the same label and with probability $q$ otherwise, and we set $(p, q) = (0.02, 0.002)$ in the source domain, which yields a strongly homophilic graph.

To induce FS only, we first generate the target-domain feature and labels with the same procedure as in the source domain. Then we rotate all target-domain features by $30°$ or $60°$ around the origin, which yields $P_S(\boldsymbol{X}) \neq P_T(\boldsymbol{X})$. This orthogonal transformation preserves all pairwise distances, so the clustering labels remain unchanged, and we keep the same SBM parameters $(p, q)$ for edge generation. Consequently, $P_S(\boldsymbol{A} \mid \boldsymbol{X}) = P_T(\boldsymbol{A} \mid \boldsymbol{X})$ holds. No rotation, a $30°$ rotation, and a $60°$ rotation correspond to no FS, mild FS, and severe FS, respectively.

To induce SS only, we generate the target-domain features and labels in the same way as the source domain, and only change the SBM parameters $(p, q)$ in the target domain. These settings change the relative frequency of inter-class edges, making the target graphs with different homophily. Since labels are determined by features and edges are generated from labels, this change alters both the structure and the feature-conditional connectivity, so $P_S(\boldsymbol{A}) \neq P_T(\boldsymbol{A})$ and $P_S(\boldsymbol{A} \mid \boldsymbol{X}) \neq P_T(\boldsymbol{A} \mid \boldsymbol{X})$ while $P_S(\boldsymbol{X}) = P_T(\boldsymbol{X})$. The choice of $(p, q) = (0.02, 0.002), (0.02, 0.010), (0.015, 0.016)$ in the target domain represents no SS, mild SS and severe SS, respectively. To induce both FS and SS, we simultaneously rotate the features and modify $(p, q)$ in the target domain.

The experimental results on the synthetic datasets are presented in Table 1. For all tasks, accuracy is reported as the mean and 1-sigma standard deviation over 5 independent runs. We observe that our DACS model significantly outperforms all baseline methods, particularly under severe shift conditions and when both FS and SS are present. PairAlign achieves the second-best performance in the

two SS-only cases (columns 4 and 5) by reconstructing node connection probabilities based on class labels. However, when FS is introduced, its performance drops significantly, especially at a rotation degree of $60°$, as PairAlign lacks the ability to effectively handle FS. Moreover, under the severe-FS-only case (column 3), both our DACS method and GAA achieve over 90% accuracy. However, when mild SS or severe SS is introduced (columns 8–9), GAA's performance drops dramatically, while DACS outperforms it by 12.05% and 13.25%, likely since GAA lacks a module specifically designed to address SS. Note that several methods achieved impressive performance in FS-only and SS-only cases, yet experienced significant performance degradation when different types of shifts exist. These results highlight the effectiveness of our dual alignment approach in addressing various CS scenarios by disentangling CS into its feature and structure components. To validate the robustness of our DACS, we evaluate its performance under noisy FS, noisy FCSS, and reduced feature separability in Appendix C.5.

Table 1: Performance on synthetic datasets.

| Rotation Degree | Rotation $30°$ | Rotation $60°$ | No Feature Shift | | Rotation $30°$ | | Rotation $60°$ | |
|---|---|---|---|---|---|---|---|---|
| Probability $(p, q)$ | No Structure Shift | | (0.02, 0.01) | (0.015, 0.016) | (0.02, 0.01) | (0.015, 0.016) | (0.02, 0.01) | (0.015, 0.016) |
| UDAGCN | 86.46±1.65 | 80.34±1.90 | 86.72±1.62 | 86.71±0.93 | 85.53±0.29 | 79.91±1.95 | 76.23±1.17 | 72.09±0.86 |
| ASN | 93.29±0.13 | 86.69±1.10 | 96.28±0.44 | 93.45±0.22 | 91.17±2.13 | 89.42±1.64 | 84.54±0.69 | 80.77±1.52 |
| AdaGCN | 86.71±1.62 | 81.43±1.07 | 86.73±1.65 | 86.69±1.62 | 86.60±1.64 | 86.44±1.02 | 80.72±1.64 | 78.84±3.23 |
| CWGCN | 93.24±0.26 | 90.08±1.15 | 96.87±0.62 | 93.46±1.30 | 87.01±1.28 | 86.95±1.24 | 85.14±1.59 | 79.83±2.84 |
| JHGDA | 80.20±2.65 | 78.38±2.54 | 81.57±2.37 | 79.94±2.64 | 79.91±2.51 | 78.38±3.15 | 74.15±2.23 | 73.65±2.47 |
| GAA | 92.53±1.57 | 90.69±1.48 | 89.07±1.47 | 86.59±1.88 | 85.11±1.82 | 83.63±2.02 | 81.50±2.56 | 79.93±1.43 |
| StruRW | 85.29±1.60 | 82.45±2.14 | 86.07±1.67 | 84.81±1.78 | 84.47±1.74 | 82.58±1.32 | 81.64±2.04 | 80.78±2.38 |
| PairAlign | 96.10±0.22 | 79.43±1.69 | 98.93±0.09 | 97.91±0.13 | 95.27±0.24 | 94.93±0.27 | 75.12±0.57 | 72.45±1.48 |
| DACS | **99.81±0.15** | **94.93±0.26** | **100±0.00** | **99.82±0.12** | **99.23±0.14** | **98.74±0.19** | **93.55±0.78** | **93.18±0.89** |

The best and second-best performances are marked as **bold** and underline, respectively.

**Real-world Experiments.** We conduct two GDA tasks between two paper citation networks `DBLPv8` and `ACMv9` represented by `D` and `A` chosen from `DBLP` after year 2010 and `ACM` datasets within years 2000-2010 (Tang et al., 2008; Wu et al., 2022). Moreover, `Airport` dataset (Ribeiro et al., 2017) includes three airport traffic networks `Brazil`, `Europe` and the `USA`, represented by `B`, `E` and `U`. `ArXiv` dataset (Hu et al., 2020) is a citation network in which papers are categorized into different subject areas. The source and target data consist of papers published in two non-overlapping time periods. On `ArXiv`, we omit the baselines that are out-of-memory (OOM) for all tasks. To handle its large scale efficiently, our DACS algorithm applies $k$-nearest neighbour strategy for each node to choose the node pairs for estimating the weights with the computation complexity $O((|\mathcal{V}_S| + |\mathcal{V}_T|)kLd + |\mathcal{E}_S|Ld)$. We set $k = 100$ so that the total number of node pairs, $|\mathcal{V}| \cdot k$, is sufficiently large to ensure accurate and stable weight estimation. Detailed description of these datasets and complexity analysis can be found in Appendix C.1 and D, respectively.

The experimental results on the real-world datasets are shown in Tables 2 and 3. Notably, our DACS method achieves superior performance across almost all GDA tasks, with maximally 7.84% of improvements for `A-D`. These results underscore the effectiveness of jointly considering and addressing both FS and FCSS in tackling GDA challenges on different real-world datasets. Model analysis on edge weight estimation and alignment tools, parameter analysis for the reweighting parameter $\lambda$ and the number of intervals $J$, and visualization for the ablation study can be found in Appendix C.2-C.4.

Table 2: Performance on `DBLP`/`ACM` and `Airport` datasets.

| Models | DBLP/ACM | | Airport | | | | | |
|---|---|---|---|---|---|---|---|---|
| | A-D | D-A | U-E | E-U | B-E | E-B | B-U | U-B |
| UDAGCN | 68.86±1.08 | 63.91±1.16 | 48.87±1.45 | 43.41±0.52 | 50.77±0.70 | 47.62±1.13 | 49.78±0.55 | 61.22±1.25 |
| ASN | 72.70±0.38 | 71.62±0.78 | 46.45±0.34 | 46.25±2.65 | 49.62±1.04 | 59.03±1.56 | 49.86±2.37 | 51.91±0.89 |
| AdaGCN | 68.59±1.12 | 66.75±1.59 | 43.64±0.82 | 45.57±0.41 | 48.52±0.63 | 56.17±1.22 | 50.30±1.45 | 47.24±1.20 |
| CWGCN | 76.74±0.11 | 72.75±0.45 | 47.03±0.85 | 45.74±0.95 | 48.77±0.52 | 63.96±0.89 | 49.82±1.46 | 62.29±0.78 |
| JHGDA | 75.58±1.65 | 73.22±0.68 | 50.75±3.01 | 52.27±6.25 | 56.64±1.14 | **73.13±1.37** | 50.20±0.54 | 69.27±3.32 |
| GAA | 78.37±7.43 | 74.76±1.57 | 55.96±3.69 | 53.33±1.24 | 56.64±0.75 | 68.30±2.08 | 53.22±1.25 | 69.81±4.92 |
| StruRW | 70.19±2.10 | 66.57±0.42 | 53.77±0.98 | 49.67±2.88 | 56.06±1.31 | 65.65±0.15 | 52.19±2.01 | 62.84±0.11 |
| PairAlign | 75.24±1.06 | 74.77±1.57 | 55.39±0.94 | 54.28±2.07 | 55.72±0.97 | 52.90±1.48 | 52.78±1.62 | 67.86±0.50 |
| DACS | **86.21±0.24** | **79.14±0.31** | **56.58±0.36** | **55.72±0.08** | **57.39±0.15** | 70.76±0.19 | **54.09±0.13** | **74.34±0.30** |

The best and second-best performances are marked as **bold** and underline, respectively.

**Ablation Analysis.** We evaluate three variants of our DACS model to examine how its components handle GDA. These variants include DANN (Ganin et al., 2016), a classic adversarial alignment

Table 3: Performance on `Arxiv` dataset.

| Models | 1950-2007 | | 1950-2009 | | 1950-2011 | |
|---|---|---|---|---|---|---|
| | 2014-2016 | 2016-2018 | 2014-2016 | 2016-2018 | 2014-2016 | 2016-2018 |
| UDAGCN | $38.10 \pm 1.62$ | OOM | $42.85 \pm 2.09$ | OOM | $\underline{53.13 \pm 0.31}$ | OOM |
| AdaGCN | $37.69 \pm 1.19$ | $43.01 \pm 1.87$ | $\underline{45.50 \pm 0.68}$ | $47.53 \pm 1.12$ | $51.37 \pm 1.12$ | $53.34 \pm 1.60$ |
| CWGCN | $32.53 \pm 0.52$ | $35.10 \pm 1.95$ | $42.05 \pm 0.37$ | $36.97 \pm 2.31$ | $46.08 \pm 0.99$ | $41.56 \pm 0.59$ |
| StruRW | $\underline{38.56 \pm 0.77}$ | $37.17 \pm 2.75$ | $43.55 \pm 2.37$ | $\underline{43.55 \pm 2.37}$ | $53.19 \pm 0.45$ | $\underline{53.64 \pm 0.65}$ |
| PairAlign | $39.98 \pm 0.77$ | $\underline{41.14 \pm 2.07}$ | $44.60 \pm 0.42$ | $44.43 \pm 0.34$ | $53.75 \pm 0.48$ | $52.83 \pm 0.98$ |
| DACS | $\mathbf{42.94 \pm 2.25}$ | $\mathbf{45.33 \pm 4.26}$ | $\mathbf{45.87 \pm 0.97}$ | $\mathbf{50.35 \pm 2.03}$ | $\mathbf{54.80 \pm 0.75}$ | $\mathbf{55.01 \pm 1.68}$ |

The best and second-best performances are marked as **bold** and underline, respectively.

method adapted to GNN encoded representations for GDA, which minimizes Eq. (3) with weighted $\boldsymbol{H}_{\hat{\boldsymbol{w}},u}^{L}$ replaced by unweighted $\boldsymbol{H}_u^L$; DANN+FS, which extends DANN by incorporating MLPs with adversarial loss $\mathcal{R}_{\text{FA}}$ in Eq. (2) to address FS; DANN+FCSS, which reweights the representation of each GNN layer in the source domain and minimize the loss in Eq. (3) with $\phi^0(\boldsymbol{X}) = \boldsymbol{X}$ to address FCSS. These ablation models and their ablation results are shown in Table 4 and 5, respectively.

We observe that both DANN+FS and DANN+FCSS outperform the baseline DANN, indicating that the modules introduced in Sections 5.1 and 5.2 are effective in individually addressing FS and FCSS. More importantly, our full DACS model (DANN+FS+FCSS in Eq. (5)) significantly outperforms both DANN+FS and DANN+FCSS, demonstrating the benefits of combining the two modules to jointly tackle FS and FCSS.

Table 4: Ablation Models.

| Models | $\mathcal{R}_{\text{CE}}$ | $\mathcal{R}_{\text{CA}}$ | | $\mathcal{R}_{\text{FA}}$ |
|---|---|---|---|---|
| | | Unweighted | Weighted | |
| DANN | ✔ | ✔ | – | ✗ |
| DANN+FS | ✔ | ✔ | – | ✔ |
| DANN+FCSS | ✔ | – | ✔ | ✗ |
| DACS | ✔ | – | ✔ | ✔ |

Table 5: Ablation results on the variants of the proposed DACS method.

| Models | DBLP/ACM | | Airport | | | | | |
|---|---|---|---|---|---|---|---|---|
| | A-D | D-A | U-E | E-U | B-E | E-B | B-U | U-B |
| DANN | 64.64±1.36 | 54.36±0.14 | 49.33±0.24 | 47.76±0.17 | 50.99±0.10 | 67.54±0.30 | 50.62±0.11 | 65.47±0.51 |
| DANN+FS | 75.10±0.59 | 64.15±0.32 | 55.71±0.23 | 52.64±0.14 | 56.77±0.22 | 69.81±0.65 | 53.45±0.25 | 71.88±0.61 |
| DANN+FCSS | 71.91±0.89 | 62.44±0.26 | 51.30±0.16 | 49.18±0.14 | 55.65±0.06 | 69.62±0.32 | 52.12±0.42 | 71.13±0.51 |
| DACS | **86.21±0.24** | **79.14±0.19** | **56.58±0.36** | **55.72±0.41** | **57.39±0.15** | **70.75±0.24** | **54.08±0.12** | **74.34±0.32** |

# 7 CONCLUSION AND FUTURE WORK

We addressed the challenge of CS in GDA by decomposing it into two complementary components: FS and FCSS. Building on this perspective, we proposed DACS, a dual-alignment framework that integrates three key modules: adversarial training to align feature distributions, layer-wise reweighting to adaptively correct structural dependencies, and final-layer alignment to unify the learned representations. Extensive experiments on benchmark datasets demonstrated that our DACS method substantially reduces domain discrepancies and consistently improves cross-domain generalization in GDA tasks.

While our analysis and algorithm are developed under the static graph setting, extending the framework to dynamic graphs—where both node features and graph structures evolve over time—could further broaden its applicability. In such scenarios, incorporating temporal dynamics and structural revolution into the framework would enable it to better capture realistic graph processes, such as evolving social networks, communication systems, and financial transaction graphs, thereby enhancing its utility in a wider range of real-world applications.

Recent studies in multi-domain graph generalization (MDGG) (Zhao et al., 2024a; Wang et al., 2025) investigate transferring knowledge across highly heterogeneous graphs, including shifts between homophilic and heterophilic structures and even mismatched feature or label semantics. In contrast, our work operates within the classical GDA setting, where the source and target graphs are assumed to share a highly relevant and compatible underlying distribution, enabling effective transfer of structural and semantic information. Although DACS does not aim to address the full spectrum of heterogeneity considered in MDGG, its structural correction mechanism can be naturally integrated with graph rewiring or graph structure learning to support more general cross-domain scenarios. Exploring this broader connection represents a promising direction for future work.

## ETHICS STATEMENT

This work makes use of publicly available datasets and models. No private or sensitive data is involved, and no harmful content is included. Therefore, we believe this paper does not raise any ethical concerns.

## REPRODUCIBILITY STATEMENT

Implementation details for our proposed algorithms and complete descriptions of all datasets used are provided in Appendix C.1. The corresponding algorithm codes and data processing scripts can be accessed via the anonymous link `https://anonymous.4open.science/r/DualAlign`. Full proofs of the theoretical claims are included in Appendix A.

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

This appendix provides supplementary material to the main paper, including technical details, theoretical proofs, and extended experiments. Appendix A presents the formal proofs of our theoretical results—most notably the target risk bound and the conditional expectation alignment theorem—which together establish the foundations of the proposed dual-alignment framework DACS. Appendix B then describes the DACS algorithm in detail, offering step-by-step explanations of the training objectives, optimization procedures, and the overall algorithmic pipeline. Appendix C reports additional experimental studies, covering dataset descriptions, implementation details, model analyses, parameter sensitivity evaluations, and ablation experiments. These results further validate the effectiveness of our DACS method across diverse scenarios. Taken together, the appendices reinforce both the theoretical soundness and the empirical advantages of our approach, ensuring clarity, reproducibility, and completeness of the work.

## A  PROOFS

*Proof of Theorem 3.1.* Similar to the definition of $r_S(\boldsymbol{H}^1, \boldsymbol{H}^L)$, we define the conditional target risk as

$$r_T(\boldsymbol{H}^1, \boldsymbol{H}^L) := \mathbb{E}_{P_T}\big(\|g(\boldsymbol{H}^L) - f_T(\boldsymbol{X}, \boldsymbol{A})\|_1 \mid \boldsymbol{H}^1, \boldsymbol{H}^L\big).$$

Moreover, the marginal target distribution of the initial representation is defined as

$$dP_T(\boldsymbol{H}^1) := \int_{\boldsymbol{X}:\phi^0(\boldsymbol{X})=\boldsymbol{H}^1} dP_T(\boldsymbol{X}),$$

and the conditional final representation distribution given the input one is

$$dP_T(\boldsymbol{H}^L \mid \boldsymbol{H}^1) := \int_{\boldsymbol{A}:\phi(\boldsymbol{H}^1, \boldsymbol{A})=\boldsymbol{H}^L} dP_T(\boldsymbol{A} \mid \boldsymbol{H}^1).$$

For the composite classifier $g \circ \phi$, we first apply the tower rule to obtain

$$\mathcal{R}_T(g \circ \phi) = \mathcal{R}_S(g \circ \phi) + \mathcal{R}_T(g \circ \phi) - \mathcal{R}_S(g \circ \phi)$$
$$= \mathcal{R}_S(g \circ \phi) + \mathbb{E}_{P_T}(\|g \circ \phi(\boldsymbol{X}, \boldsymbol{A}) - f_T(\boldsymbol{X}, \boldsymbol{A})\|_1) - \mathbb{E}_{P_S}(\|g \circ \phi(\boldsymbol{X}, \boldsymbol{A}) - f_S(\boldsymbol{X}, \boldsymbol{A})\|_1)$$
$$= \mathcal{R}_S(g \circ \phi) + \mathbb{E}_{P_T}(\mathbb{E}_{P_T}(\|g(\boldsymbol{H}^L) - f_T(\boldsymbol{X}, \boldsymbol{A})\|_1 \mid \boldsymbol{H}^1))$$
$$\quad - \mathbb{E}_{P_S}(\mathbb{E}_{P_S}(\|g(\boldsymbol{H}^L) - f_S(\boldsymbol{X}, \boldsymbol{A})\|_1 \mid \boldsymbol{H}^1))$$
$$= \mathcal{R}_S(g \circ \phi) + \mathbb{E}_{P_T}(r_T(\boldsymbol{H}^1)) - \mathbb{E}_{P_S}(r_S(\boldsymbol{H}^1))$$
$$= \mathcal{R}_S(g \circ \phi) + \mathbb{E}_{P_T}(r_T(\boldsymbol{H}^1) - r_S(\boldsymbol{H}^1)) + (\mathbb{E}_{P_T}(r_S(\boldsymbol{H}^1)) - \mathbb{E}_{P_S}(r_S(\boldsymbol{H}^1)))$$
$$= \mathcal{R}_S(g \circ \phi) + \mathbb{E}_{P_T}(r_T(\boldsymbol{H}^1) - r_S(\boldsymbol{H}^1)) + \int_{\mathcal{H}^1} r_S(\boldsymbol{H}^1)(dP_T(\boldsymbol{H}^1) - dP_S(\boldsymbol{H}^1))$$
$$\leq \mathcal{R}_S(g \circ \phi) + \mathbb{E}_{P_T}(r_T(\boldsymbol{H}^1) - r_S(\boldsymbol{H}^1)) + \int_{\mathcal{H}^1} r_S(\boldsymbol{H}^1)|dP_T(\boldsymbol{H}^1) - dP_S(\boldsymbol{H}^1)|. \qquad (6)$$

Next, for the term $r_S(\boldsymbol{H}^1)$, applying the tower rule gives

$$r_S(\boldsymbol{H}^1) = \mathbb{E}_{P_S}\big(\|g(\boldsymbol{H}^L) - f_S(\boldsymbol{X}, \boldsymbol{A})\|_1 \mid \boldsymbol{H}^1\big)$$
$$= \mathbb{E}_{P_S}\big(\mathbb{E}_{P_S}\big(\|g(\boldsymbol{H}^L) - f_S(\boldsymbol{X}, \boldsymbol{A})\|_1 \mid \boldsymbol{H}^L, \boldsymbol{H}^1\big) \mid \boldsymbol{H}^1\big)$$
$$=: \mathbb{E}_{P_S}\big(r_S(\boldsymbol{H}^1, \boldsymbol{H}^L) \mid \boldsymbol{H}^1\big).$$

Thus we have

$$r_T(\boldsymbol{H}^1) - r_S(\boldsymbol{H}^1) = \mathbb{E}_{P_T}(r_T(\boldsymbol{H}^1, \boldsymbol{H}^L) \mid \boldsymbol{H}^1) - \mathbb{E}_{P_S}(r_S(\boldsymbol{H}^1, \boldsymbol{H}^L) \mid \boldsymbol{H}^1)$$
$$= \mathbb{E}_{P_T}(r_T(\boldsymbol{H}^1, \boldsymbol{H}^L) \mid \boldsymbol{H}^1) - \mathbb{E}_{P_T}(r_S(\boldsymbol{H}^1, \boldsymbol{H}^L) \mid \boldsymbol{H}^1)$$
$$\quad + \mathbb{E}_{P_T}(r_S(\boldsymbol{H}^1, \boldsymbol{H}^L) \mid \boldsymbol{H}^1) - \mathbb{E}_{P_S}(r_S(\boldsymbol{H}^1, \boldsymbol{H}^L) \mid \boldsymbol{H}^1)$$
$$= \mathbb{E}_{P_T}\big(r_T(\boldsymbol{H}^1, \boldsymbol{H}^L) - r_S(\boldsymbol{H}^1, \boldsymbol{H}^L) \mid \boldsymbol{H}^1\big)$$
$$\quad + \mathbb{E}_{P_T}\left(\int r_S(\boldsymbol{H}^1, \boldsymbol{H}^L)\big(d\mathrm{P}_T(\boldsymbol{H}^L \mid \boldsymbol{H}^1) - d\mathrm{P}_S(\boldsymbol{H}^L \mid \boldsymbol{H}^1)\big)\right)$$
$$\leq \mathbb{E}_{P_T}\big(|r_T(\boldsymbol{H}^1, \boldsymbol{H}^L) - r_S(\boldsymbol{H}^1, \boldsymbol{H}^L)| \mid \boldsymbol{H}^1\big)$$
$$\quad + \int r_S(\boldsymbol{H}^1, \boldsymbol{H}^L) \cdot \big|d\mathrm{P}_T(\boldsymbol{H}^L \mid \boldsymbol{H}^1) - d\mathrm{P}_S(\boldsymbol{H}^L \mid \boldsymbol{H}^1)\big|. \qquad (7)$$

Plugging this result into Eq. (6) yields

$$\mathcal{R}_T(g \circ \phi) \leq \mathcal{R}_S(g \circ \phi) + \mathbb{E}_{P_T}|r_T(\boldsymbol{H}^1, \boldsymbol{H}^L) - r_S(\boldsymbol{H}^1, \boldsymbol{H}^L)|$$

$$+ \int_{\mathcal{H}^1} r_S(\boldsymbol{H}^1) \cdot |d\,\mathrm{P}_T(\boldsymbol{H}^1) - d\,\mathrm{P}_S(\boldsymbol{H}^1)|$$

$$+ \mathbb{E}_{P_T}\left(\int_{\mathcal{H}^L} r_S(\boldsymbol{H}^1, \boldsymbol{H}^L) \cdot |d\,\mathrm{P}_T(\boldsymbol{H}^L \mid \boldsymbol{H}^1) - d\,\mathrm{P}_S(\boldsymbol{H}^L \mid \boldsymbol{H}^1)|\right). \quad (8)$$

Since domain adaptation is solved by training a shared classifier on the aligned representation space $\mathcal{H}^L$, i.e.,

$$f_S(\boldsymbol{X}, \boldsymbol{A}) = g^*(\boldsymbol{H}^L) = f_T(\boldsymbol{X}, \boldsymbol{A}),$$

where $g^*$ is denoted as the ground-truth labeling function on the representation space of $\boldsymbol{H}^L$, the second term in the right-hand side of Eq. (8) becomes

$$|r_T(\boldsymbol{H}^1, \boldsymbol{H}^L) - r_S(\boldsymbol{H}^1, \boldsymbol{H}^L)|$$
$$= \left|\mathbb{E}_{P_T}\left(\|g(\boldsymbol{H}^L) - g^*(\boldsymbol{H}^L)\|_1 \mid \boldsymbol{H}^1, \boldsymbol{H}^L\right) - \mathbb{E}_{P_S}\left(\|g(\boldsymbol{H}^L) - g^*(\boldsymbol{H}^L)\|_1 \mid \boldsymbol{H}^1, \boldsymbol{H}^L\right)\right|$$
$$= \|g(\boldsymbol{H}^L) - g^*(\boldsymbol{H}^L)\|_1 - \|g(\boldsymbol{H}^L) - g^*(\boldsymbol{H}^L)\|_1 = 0.$$

This completes the proof. $\qquad\square$

Next, we establish the following result.

**Proposition A.1.** *If the feature-conditional adjacency distributions are aligned across domains, i.e.,*

$$P_S(\boldsymbol{A} \mid \boldsymbol{X}) = P_T(\boldsymbol{A} \mid \boldsymbol{X}),$$

*then the conditional distributions of the final-layer representations given the initial representations are also aligned:*

$$P_S(\boldsymbol{H}^L \mid \boldsymbol{H}^1) = P_T(\boldsymbol{H}^L \mid \boldsymbol{H}^1).$$

*Proof of Proposition A.1.* Since $P_S(\boldsymbol{A} \mid \boldsymbol{X}) = P_T(\boldsymbol{A} \mid \boldsymbol{X})$, we have

$$P_S(\boldsymbol{A} \mid \boldsymbol{H}^1) = P_T(\boldsymbol{A} \mid \boldsymbol{H}^1). \quad (9)$$

By the law of total probability, we begin with the following expression:

$$P(\boldsymbol{H}^L \mid \boldsymbol{H}^1) = P(\phi(\boldsymbol{H}^1, \boldsymbol{A}) \mid \boldsymbol{H}^1)$$
$$= \sum_{\boldsymbol{a} \in \mathcal{A}} P(\phi(\boldsymbol{H}^1, \boldsymbol{A}), \boldsymbol{A} = \boldsymbol{a} \mid \boldsymbol{H}^1)$$
$$= \sum_{\boldsymbol{a} \in \mathcal{A}} P(\phi(\boldsymbol{H}^1, \boldsymbol{A}) \mid \boldsymbol{A} = \boldsymbol{a}, \boldsymbol{H}^1) P(\boldsymbol{A} = \boldsymbol{a} \mid \boldsymbol{H}^1). \quad (10)$$

Since $\phi(\boldsymbol{H}^1, \boldsymbol{A})$ is a function of both $\boldsymbol{A}$ and $\boldsymbol{H}^1$, we have the equality

$$P_S(\phi(\boldsymbol{H}^1, \boldsymbol{A}) \mid \boldsymbol{A} = \boldsymbol{a}, \boldsymbol{H}^1) = P_T(\phi(\boldsymbol{H}^1, \boldsymbol{A}) \mid \boldsymbol{A} = \boldsymbol{a}, \boldsymbol{H}^1),$$

This together with Eq. (9) and Eq. (10) yields

$$P_S(\boldsymbol{H}^L \mid \boldsymbol{H}^1) = P_T(\boldsymbol{H}^L \mid \boldsymbol{H}^1),$$

which completes the proof. $\qquad\square$

*Proof of Theorem 4.1.* By applying the tower rule, we obtain

$$\mathbb{E}_{P_T}(\boldsymbol{A}\boldsymbol{H}^\ell \mid \boldsymbol{H}^1) = \mathbb{E}_{P_T}(\mathbb{E}_{P_T}(\boldsymbol{A}\boldsymbol{H}^\ell \mid \boldsymbol{H}^\ell, \boldsymbol{H}^1) \mid \boldsymbol{H}^1). \quad (11)$$

Moreover, we have

$$\mathbb{E}_{P_T}(\boldsymbol{A}\boldsymbol{H}^\ell \mid \boldsymbol{H}^\ell, \boldsymbol{H}^1) = \mathbb{E}_{P_S}\left(\frac{P_T(\boldsymbol{H}^\ell, \boldsymbol{A} \mid \boldsymbol{H}^\ell, \boldsymbol{H}^1)}{P_S(\boldsymbol{H}^\ell, \boldsymbol{A} \mid \boldsymbol{H}^\ell, \boldsymbol{H}^1)} \cdot \boldsymbol{A}\boldsymbol{H}^\ell \,\bigg|\, \boldsymbol{H}^\ell, \boldsymbol{H}^1\right)$$

$$= \mathbb{E}_{P_S}\left(\frac{P_T(\boldsymbol{A} \mid \boldsymbol{H}^\ell, \boldsymbol{H}^1)}{P_S(\boldsymbol{A} \mid \boldsymbol{H}^\ell, \boldsymbol{H}^1)} \cdot \boldsymbol{A} \,\bigg|\, \boldsymbol{H}^\ell, \boldsymbol{H}^1\right) \cdot \boldsymbol{H}^\ell.$$

This together with Eq. (11) yields that

$$\mathbb{E}_{P_T}(\boldsymbol{A}\boldsymbol{H}^\ell \,|\, \boldsymbol{H}^1) = \mathbb{E}_{P_T}\left(\mathbb{E}_{P_S}\left(\frac{P_T(\boldsymbol{A}\,|\,\boldsymbol{H}^\ell,\boldsymbol{H}^1)}{P_S(\boldsymbol{A}\,|\,\boldsymbol{H}^\ell,\boldsymbol{H}^1)}\cdot \boldsymbol{A}\,\middle|\,\boldsymbol{H}^\ell,\boldsymbol{H}^1\right)\cdot \boldsymbol{H}^\ell\,\middle|\,\boldsymbol{H}^1\right). \tag{12}$$

By using the tower rule again, we can write

$$\mathbb{E}_{P_S}(w^\ell \boldsymbol{A}\boldsymbol{H}_w^\ell \,|\, \boldsymbol{H}^1) = \mathbb{E}_{P_S}\left(\frac{P_T(\boldsymbol{A}\,|\,\boldsymbol{H}_w^\ell,\boldsymbol{H}^1)}{P_S(\boldsymbol{A}\,|\,\boldsymbol{H}_w^\ell,\boldsymbol{H}^1)}\cdot \boldsymbol{A}\boldsymbol{H}_w^\ell\,\middle|\,\boldsymbol{H}^1\right)$$

$$= \mathbb{E}_{P_S}\left(\mathbb{E}_{P_S}\left(\frac{P_T(\boldsymbol{A}\,|\,\boldsymbol{H}_w^\ell,\boldsymbol{H}^1)}{P_S(\boldsymbol{A}\,|\,\boldsymbol{H}_w^\ell,\boldsymbol{H}^1)}\cdot \boldsymbol{A}\boldsymbol{H}_w^\ell\,\middle|\,\boldsymbol{H}_w^\ell,\boldsymbol{H}^1\right)\,\middle|\,\boldsymbol{H}^1\right)$$

$$= \mathbb{E}_{P_S}\left(\mathbb{E}_{P_S}\left(\frac{P_T(\boldsymbol{A}\,|\,\boldsymbol{H}_w^\ell,\boldsymbol{H}^1)}{P_S(\boldsymbol{A}\,|\,\boldsymbol{H}_w^\ell,\boldsymbol{H}^1)}\cdot \boldsymbol{A}\,\middle|\,\boldsymbol{H}_w^\ell,\boldsymbol{H}^1\right)\boldsymbol{H}_w^\ell\,\middle|\,\boldsymbol{H}^1\right). \tag{13}$$

Let us denote

$$f(\boldsymbol{H}^\ell,\boldsymbol{H}^1) := \mathbb{E}_{P_S}\left(\frac{P_T(\boldsymbol{A}\,|\,\boldsymbol{H}^\ell,\boldsymbol{H}^1)}{P_S(\boldsymbol{A}\,|\,\boldsymbol{H}^\ell,\boldsymbol{H}^1)}\cdot \boldsymbol{A}\,\middle|\,\boldsymbol{H}^\ell,\boldsymbol{H}^1\right)\cdot \boldsymbol{H}^\ell.$$

By applying Eq. (12) and Eq. (13), we obtain

$$\mathbb{E}_{P_T}(\boldsymbol{A}\boldsymbol{H}^\ell \,|\, \boldsymbol{H}^1) = \mathbb{E}_{P_T}\big[f(\boldsymbol{H}^\ell,\boldsymbol{H}^1)\,|\,\boldsymbol{H}^1\big]$$

and

$$\mathbb{E}_{P_S}(w^\ell \boldsymbol{A}\boldsymbol{H}_w^\ell \,|\, \boldsymbol{H}^1) = \mathbb{E}_{P_S}\big[f(\boldsymbol{H}_w^\ell,\boldsymbol{H}^1)\,|\,\boldsymbol{H}^1\big].$$

Since for any encoder $\phi^k$ with $k \in [\ell-1]$, the conditional distributions of $\boldsymbol{H}^\ell$ and $\boldsymbol{H}_w^\ell$ given $\boldsymbol{H}^1$ are identical, i.e.,

$$\mathbb{E}_{P_T}(\boldsymbol{H}^\ell \,|\, \boldsymbol{H}^1) = \mathbb{E}_{P_S}(\boldsymbol{H}_w^\ell \,|\, \boldsymbol{H}^1),$$

it follows that

$$P_T(\boldsymbol{H}^\ell \,|\, \boldsymbol{H}^1) = P_S(\boldsymbol{H}_w^\ell \,|\, \boldsymbol{H}^1),$$

which implies

$$\mathbb{E}_{P_T}\big[f(\boldsymbol{H}^\ell,\boldsymbol{H}^1)\,|\,\boldsymbol{H}^1\big] = \mathbb{E}_{P_S}\big[f(\boldsymbol{H}_w^\ell,\boldsymbol{H}^1)\,|\,\boldsymbol{H}^1\big].$$

Therefore, we conclude that

$$\mathbb{E}_{P_T}(\boldsymbol{A}\boldsymbol{H}^\ell \,|\, \boldsymbol{H}^1) = \mathbb{E}_{P_S}(w^\ell \boldsymbol{A}\boldsymbol{H}_w^\ell \,|\, \boldsymbol{H}^1),$$

which completes the proof. $\square$

## B  DUAL-ALIGNMENT ALGORITHM FOR COVARIATE SHIFT

---
**Algorithm 1** Dual Alignment for Covariate Shift (DACS)

---
**Input:** The source graph $\mathcal{G}_S$ with labels $\mathcal{Y}_S$; a MLP $\phi^0$; a GNN $\phi$ and a classifier $g$; the total epoch number $E$.
  **while** epoch $< E$ or not converge **do**
    Calculate the weight estimator $\widehat{w}_{uv}^\ell$.
    Update the MLP $\phi^0$, the GNN $\phi$ and a classifier $g$ by minimizing the combined training risks as in Eq. (5).
  **end while**
**Output:** The MLP $\phi^0$, GNN $\phi$ and classifier $g$.

---

## C  COMPLEMENTARY EXPERIMENTS

### C.1  DATASETS AND IMPLEMENTATION DETAILS

**Datasets.** DBLP, ACM are two paper citation networks obtained from different original sources (Tang et al., 2008). We choose DBLPv8 and ACMv9 that contain papers published in different periods and use the processed versions from Wu et al. (2022). DBLPv8 is after year 2010 and ACMv9 is collected between years 2000 and 2010. In these networks, each node represents a paper, and each edge indicates a citation between two papers. The features of a node are represented by a bag-of-words vector, indicating keywords extracted from the title of the corresponding paper. We classify the paper in the networks into one of the following six categories according to its research topics, including "Database", "Data Mining", "Artificial Intelligence", "Computer Vision", "Information Security", and "High Performance Computing". Brazil, Europe and the USA are airport traffic networks from the Airport datasets (Ribeiro et al., 2017). In these networks, nodes represent airports, and edges denote flight connections. Airports are labeled based on activity levels, measured by flights or passenger numbers. ArXiv (Hu et al., 2020) is a citation network of ArXiv papers, where the task is to classify papers into their respective subject areas. To investigate temporal distribution shifts, we use papers published in earlier periods as the source domain and those from later periods as the target domain. Specifically, the source data comprises papers published between 1950 and 2007/2009/2011, while the target data includes papers published during 2014–2016 and 2016–2018. In Table 6, we summarize the number of nodes, the number of edges, and the number of labels for each dataset.

We use A and D to represent the ACMv9 and DBLPv8 networks. B, E and U are used to represent Brazil, Europe, and USA respectively in the Airport dataset. In the experiments for real-world datasets, we carry out two transfer learning tasks between the citation networks and perform six cross-domain tasks for the Airport dataset. In the experiments for synthetic datasets, we investigate two situations for each of FS and SS, where the degree of shift varies between them.

Table 6: Statistics of datasets used in our experiments.

| Dataset | Domains | #Nodes | #Edges | #Labels |
|---------|---------|--------|--------|---------|
| DBLP/ACM | ACMv9 | 7,410 | 11,135 | 6 |
|         | DBLPv8 | 5,578 | 7,341 | |
| Airport | USA | 1,190 | 13,599 | 4 |
|         | Brazil | 131 | 1,038 | |
|         | Europe | 399 | 5,995 | |
| Arxiv | 1950–2007 | 4,980 | 5,849 | 40 |
|       | 1950–2009 | 9,410 | 13,179 | |
|       | 1950–2011 | 17,401 | 30,486 | |
|       | 2014–2016 | 69,499 | 232,419 | |
|       | 2016–2018 | 120,740 | 615,415 | |

**Implementation Details.** The experiments are implemented using the PyTorch platform on a workstation equipped with an Intel(R) Core(TM) i7-14700K CPU@3.40GHz and a NVIDIA GeForce RTX 4080 16GB GPU. To address FS, we use an MLP model for both the feature extractor and the domain discriminator. The number of layers in the feature extractor is selected from $\{1, 2, 3\}$, with hidden dimensions chosen from $\{32, 64, 128, 256\}$ for real-world datasets and $\{8, 16, 32, 64\}$ for synthetic datasets. The domain discriminator is implemented as a two-layer MLP with hidden dimensions selected from $\{8, 16, 32, 64\}$ across all datasets. To address FCSS, we utilize a GNN model to learn node features. The number of GNN layers is also selected from $\{1, 2, 3\}$, and its hidden dimensions are chosen from $\{32, 64, 128, 256\}$ for real-world datasets and $\{8, 16, 32, 64\}$ for synthetic datasets. The domain discriminator settings for FCSS are identical to those used for FS.

We select the learning rate in $\{0.0001, 0.001, 0.003, 0.01\}$ for the synthetic and Airport datasets, following Liu et al. (2024b). For the DBLP and ACM datasets, the learning rate is set to 0.007, as suggested in Liu et al. (2023). $\lambda$ is the reweighting ratio that depicts the level of dependence on reweighting edges, as stated in Section 5.2, where $\lambda = 1$ stands for the case where GNN completely

adopts the reweighted graph and $\lambda = 0$ corresponds to using the original graph. When estimating $w_{uv}^{\ell}$: for DBLP, ACM, and Airport, we keep all pairs of node distances; for Arxiv, we select the top-$k$ largest distances for each node to reduce computation. In practice, $k$ is set as 100 for simplicity. The setting of hyperparameters $\gamma_{\text{FA}}$ and $\gamma_{\text{CA}}$ follows the schedule: $\min\{2/(1 + e^{-10p}) - 1, 0.1\}$, where $p$ changes from 0 to 1 during the training process, as described in Ganin et al. (2016). For all datasets, we use the source graph for training. 20% of node labels in the target graph are used for validation and the remaining 80% are for testing.

The total number of training epochs is set to 300. Similar to the reweighting strategy in Liu et al. (2023), we define three key parameters: the starting epoch $e$, the reweighting frequency $t$, and the reweighting ratio $\lambda$. The starting epoch determines when to begin imposing edge weights on the source graph. The reweighting frequency specifies how often the edge weights are updated. The reweighting ratio represents the balance factor between the reweighted message and the original message during feature aggregation. The search spaces for $e$, $t$ and $\lambda$ are $\{100, 150, 200, 250\}$, $\{1, 5, 10, 15\}$ and $\{0.1, 0.2, 0.3, 0.4, 0.5, 0.6, 0.7, 0.8, 0.9\}$, respectively.

## C.2 MODEL ANALYSIS

In this section, we evaluate different weight estimation strategies based on representations from different layers and different domain adaptation models used to mitigate FS and FCSS.

We first conducted experiments using four different weight estimation strategies to analyze how the choice of feature level affects weight estimation. These strategies are defined as follows: $\text{RW}(\boldsymbol{H}^L)$ reweights the edges in the source domain based on the last-layer representation, $\boldsymbol{H}^L$; $\text{RW}(\boldsymbol{H}^k)$ reweights edges using the $k$-th layer representation, $\boldsymbol{H}^k$, as stated in Section 5.2; $\text{RW}(\boldsymbol{H}^1)$ reweights based on the input to the GNN, $\boldsymbol{H}^1$; $\text{RW}(\boldsymbol{X})$ reweights using the original node features, $\boldsymbol{X}$. We observe from Figure 4(a) that $\text{RW}(\boldsymbol{X})$ and $\text{RW}(\boldsymbol{H}^1)$, which both operate at a lower level of feature abstraction, obtain less effective results. This may be because the extracted features are not sufficiently expressive for accurate weight estimation. The best performance is observed in $\text{RW}(\boldsymbol{H}^k)$. This suggests that estimating edge weights by using the representations from the same layer appears to be the most effective way, as it aligns the weight estimation process with the level of feature abstraction inherent to that layer.

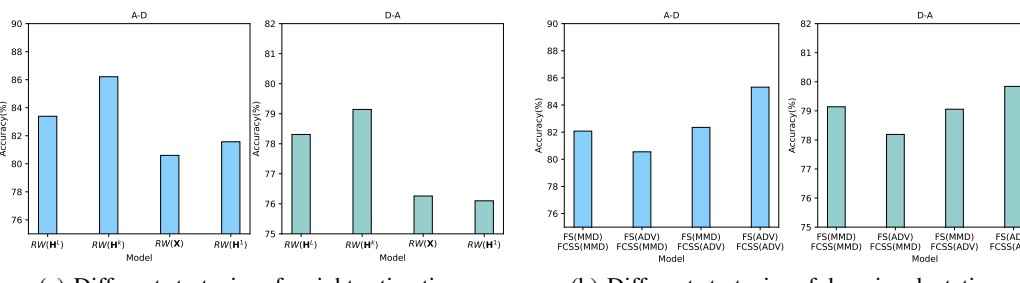

(a) Different strategies of weight estimation.   (b) Different strategies of domain adaptation.

Figure 4: Results of model analysis on reweighting and domain adaptation strategies.

As described in Section 5.2, our dual alignment approach applies adversarial alignment (ADV) to address FS and FCSS through Eq. (2) and Eq. (3), respectively. We then investigate the effect of domain adaptation strategies by exploring various combinations of alignment methods for solving FS and FCSS. Specifically, we compare pairwise combinations of maximum mean discrepancy (MMD) and ADV. As shown in Figure 4(b), applying MMD—whether to FS, FCSS, or both—leads to performance degradation compared to using ADV, which indicates that ADV is more effective than MMD in GDA in both A-D and D-A. Moreover, we observe that using the same alignment method (either MMD or ADV) for both FS and FCSS results in better or comparable performance than mixing the two. This highlights the importance of consistency in alignment strategies to solve FS and FCSS.

## C.3 PARAMETER ANALYSIS

In this part, we first analyze the influence of parameters on the performance of our dual-alignment method DACS. The following parameters are considered: *(i)* the number of MLP layers to address FS, *(ii)* the feature dimension at the final hidden layer, *(iii)* the reweighting parameter $\lambda$, and *(iv)* the number of distance intervals $J$. We omit the experiments on the number of GNN layers for addressing FCSS, as their influence on GDA performance has been well studied in prior works.

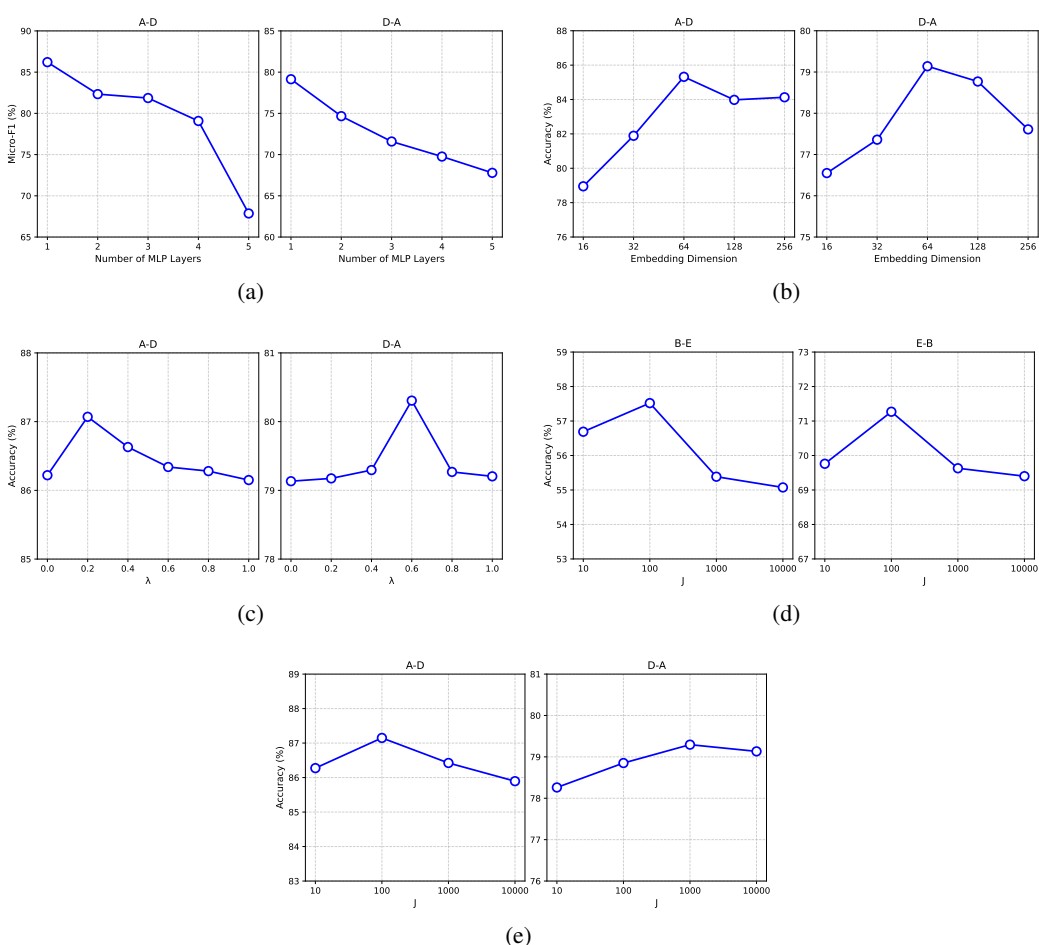

Figure 5: Results of parameter analysis.

As shown in Figure 5(a), excessively deep MLPs can degrade performance. It shows that overly aggressive feature alignment, especially at the current stage where structural information is not yet incorporated, can obscure the semantics encoded in the raw node features. Figure 5(b) shows that an embedding dimension of 64 yields the best performance in both tasks.

The impact of the reweighting coefficient $\lambda$ on GDA performance is shown in Figure 5(c). This parameter determines the proportion of edge probability ratios in the weight estimation, which affects the local aggregation process in GNNs. $\lambda = 1$ indicates full reliance on the edge probability ratios, while $\lambda = 0$ corresponds to using the original, unweighted representations. Figure 5(c) shows that across the entire range of tested $\lambda$ values, the maximum accuracy difference is only 1%, the performance curve is essentially flat, and no sharp peaks or brittle regions appear. This indicates that the model is robust to the choice of $\lambda$. In practice, $\lambda$ can also be selected according to the performance on the validation set, just as in other domain adaptation methods. Therefore, this hyperparameter does not create practical barriers when applying the method to new datasets.

The number of distance intervals $J$ determines the partitions of all pairwise distances in the target domain. For large graphs, an extremely wide range of $J$ values is required to ensure that variations in $J$ lead to meaningful differences in the estimated edge probabilities.

In Figure 5(d), we sweep $J$ over a very coarse and extremely wide range: $J \in \{10, 100, 1000, 10000\}$ on small graphs `Brazil` and `Europe`, introducing changes of up to three orders of magnitude. Even under such drastic variation, accuracy performance changes remain within $2\%$, the method does not collapse at extreme values, and both B-E and E-B tasks remain stable.

To further verify robustness, we additionally analyse $J$ on the larger `DBLP` and `ACM` dataset in Figure 5(e). On the two datasets where distance statistics are far richer, the performance of DACS is even more stable since the performance changes are only within around $1\%$, showing smaller fluctuations than on the small Airport graph. This confirms that the sensitivity concerns diminish further as the dataset size increases.

From the parameter analysis, we found that $J$ is a robust parameter across different tasks and $J = 100$ works consistently well. As a result, we fix $J = 100$ across all experiments without any dataset-specific tuning. This directly demonstrates that DACS does not require delicate hyperparameter adjustments to achieve strong performance.

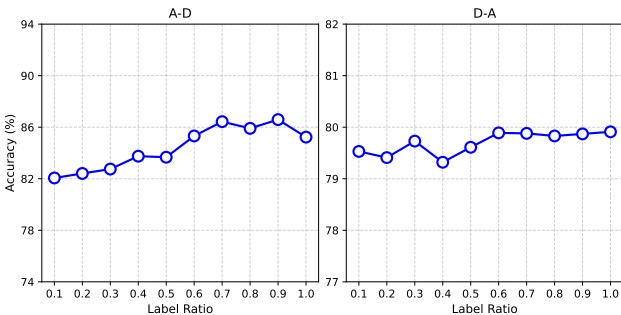

Figure 6: Performance of our DACS model under different label ratios.

In addition, we have analyzed the proportion of labeled source data used in the cross-entropy loss in Eq. (4). The ratio of training labels in the source domain can demonstrate the effectiveness of our DACS model in semi-supervised learning scenarios. We vary the ratio of labeled data involved in the classification loss from $10\%$ to $100\%$ and report the performance in Figure 6. Our DACS model exhibits relatively low sensitivity to the amount of supervision. This is reasonable as our DACS model primarily addresses covariate shift by learning the domain-invariant representation, rather than relying heavily on labels.

### C.4 Visualization for Ablation Study

We further present the visualization results for different variants in the ablation study described in Section 6. Light-colored points represent samples from the source graph, while dark-colored points are from the target graph. The performance is recorded on the `A-D` task. Three classes of nodes are randomly selected from the dataset for demonstration. We can observe that either DANN+FS or DANN+FCSS can align features from two domains. Our dual-alignment model DACS overall leads to more obvious mixing effects for nodes across domains, which shows the effectiveness of our adaptation strategies.

### C.5 Robustness Analysis under Noisy FS, Noisy FCSS, and Reduced Feature Separability

To further validate the stability of our method, we conduct a series of robustness and sensitivity experiments that introduce controlled perturbations to the source and target domains. These interventions are designed to directly reflect the two primary components of domain shift identified in our theoretical analysis: Feature Shift (FS) and Feature-Conditioned Structural Shift (FCSS). Specifi-

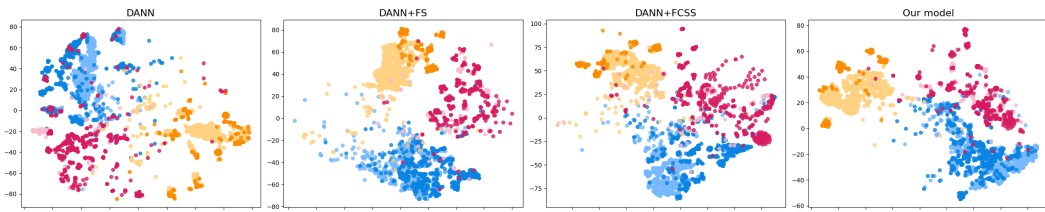

Figure 7: Visualization results of node embeddings from source and target domains for different variants of our DACS method.

cally, we examine the behavior of competing methods under noisy FS, noisy FCSS, and reduced feature separability, each of which challenges a different aspect of the alignment problem.

**Noisy FS:** We first consider the impact of noisy FS by perturbing every feature dimension with i.i.d. Gaussian noise whose standard deviation is scaled to 10%, 30%, or 50% of the original. This progressively distorts the feature representations and weakens the alignment signal available to all methods. As shown in Table 7, DACS maintains a clear margin over StruRW even as the noise level increases, indicating that our feature-alignment procedure remains effective despite substantial feature corruption.

Table 7: Performance under noisy feature shift (FS) on the A→D task.

| Method | 10% Noise | 30% Noise | 50% Noise |
|---|---|---|---|
| StruRW | 70.79±10.24 | 69.41±5.02 | 70.59±5.65 |
| **DACS** | **84.17±2.20** | **82.14±5.93** | **81.83±4.52** |

**Noisy FCSS:** We then evaluate robustness to noisy FCSS by directly perturbing the graph structure. We randomly remove 10%, 20%, or 30% of existing edges and replace them with the same number of random edges, thereby corrupting the conditional distribution $P(A \mid X)$ and injecting structural noise into the graph. Under these settings, Table 8 shows that DACS consistently outperforms StruRW across all noise levels, demonstrating that the proposed layer-wise reweighting mechanism preserves effectiveness even when the structural information becomes unreliable.

Table 8: Performance under noisy feature-conditional structure shift (FCSS) on the A→D task.

| Method | 10% Noise | 30% Noise | 50% Noise |
|---|---|---|---|
| StruRW | 67.88±3.80 | 63.80±6.50 | 60.17±7.15 |
| **DACS** | **72.69±4.39** | **66.76±4.42** | **60.99±2.62** |

**Reduced feature separability:** Finally, to assess sensitivity under more challenging data regimes, we construct an alternative synthetic dataset with reduced feature separability. The data are sampled from three Gaussian components with closer means and smaller variances,

$$P_0 = \mathcal{N}([-0.8, 0], 0.64I), \quad P_1 = \mathcal{N}([0.8, 0], 0.64I), \quad P_2 = \mathcal{N}([0, 0.8], 0.64I),$$

resulting in clusters that are substantially less distinguishable in both feature and structural space. This setting inherently makes both FS and FCSS harder to recover. Across all rotation angles and structural sparsity levels, DACS achieves the best performance, often by a large margin. Notably, even under simultaneous severe FS and FCSS, DACS continues to outperform StruRW and PairAlign, demonstrating robustness in scenarios where both alignment tasks are greatly complicated.

Together, these experiments highlight that DACS retains strong performance even under noisy, distorted, or weakly separable conditions. This consistent empirical behavior supports our core motivation for decomposing covariate shift into FS and FCSS and shows that our dual-alignment framework is intrinsically robust against a wide spectrum of real-world imperfections.

Table 9: Performance under reduced feature separability with different rotation angles and edge probabilities.

| Rotation Degree | Rotation 30° | Rotation 60° | No Feature Shift | | Rotation 30° | | Rotation 60° | |
|---|---|---|---|---|---|---|---|---|
| Probability $(p, q)$ | No Structure Shift | | $(0.02, 0.01)$ | $(0.015, 0.016)$ | $(0.02, 0.01)$ | $(0.015, 0.016)$ | $(0.02, 0.01)$ | $(0.015, 0.016)$ |
| StruRW | 99.08 | 67.39 | 77.23 | 66.65 | 59.61 | 47.15 | 50.79 | 39.64 |
| PairAlign | 96.32 | 63.00 | 92.24 | 79.59 | 72.93 | 66.56 | 55.23 | 47.62 |
| DACS | 99.98 | 85.23 | 95.09 | 87.89 | 83.78 | 77.72 | 65.91 | 52.55 |

# D    COMPLEXITY ANALYSIS

The computational bottleneck of the reweighting module lies in estimating

$$P(A_{uv} = 1 \mid d(H_{w,u}^\ell, H_{w,v}^\ell)),$$

for node pairs in both source and target domains. Rather than enumerating all $O(N^2)$ node pairs, we restrict computation to the $k$-nearest neighbors of each node. This reduces the total number of pairs to linear scale.

Let:

- $N_S, N_T$: number of nodes in the source and target graphs
- $k$: number of nearest neighbors per node
- $d$: embedding dimension
- $L$: number of GNN layers
- $|\mathcal{E}_S|$: number of edges in the source graph

## STEP 1: DISTANCE COMPUTATION FOR $k$-NEAREST NEIGHBORS

For each layer, DACS needs to:

1. compute distances for $N_S k + N_T k$ node pairs,
2. discretize the distance values into $J$ intervals, and
3. compute empirical edge probabilities within intervals.

The complexity of this step is

$$O((N_S + N_T)kd).$$

## STEP 2: REWEIGHTING SOURCE EDGES

For each source-domain edge $(u, v)$, we compute the distance $d(H_u^\ell, H_v^\ell)$ and retrieve the corresponding weight. The complexity is

$$O(|\mathcal{E}_S|d).$$

## OVERALL COMPLEXITY ACROSS $L$ LAYERS

Combining the above steps across all $L$ layers, the overall complexity is

$$O\Big((N_S + N_T)kLd + |\mathcal{E}_S|Ld\Big).$$

This is linear in the number of nodes and edges. The constants $k$, $d$, and $L$ are typically small in practice (e.g., $k = 100$, $d = 64$, $L = 2$).

## $k$-NN CONSTRUCTION PREPROCESSING

Before running DACS, the $k$ nearest neighbors for all nodes are computed as a one-time preprocessing step. Using the optimal algorithm of Ma & Li (2019), the complexity is

$$O\Big((N_S \log N_S + N_T \log N_T)k\Big),$$

which is lower than the cost of even a few GNN forward passes and can be amortized over the entire training process.

This analysis demonstrates that all computationally intensive operations are linear or near-linear in graph size, ensuring that DACS scales efficiently to large graphs.

### ACCURACY-EFFICIENCY CURVE UNDER DIFFERENT VALUES OF $k$

To complement our theoretical complexity analysis, we examine the effect of the number of nearest neighbors $k$ on runtime and accuracy using the `ArXiv` (1950–2007 $\rightarrow$ 2016–2018) task.

Figure 8 shows that as $k$ increases from 5 to 100, accuracy improves substantially, while further increasing $k$ to 1000 yields negligible gains. However, runtime and memory usage grow sharply for large $k$ (500–1000). These observations indicate that $k = 100$ achieves an excellent tradeoff between accuracy and efficiency, motivating its use in all experiments.

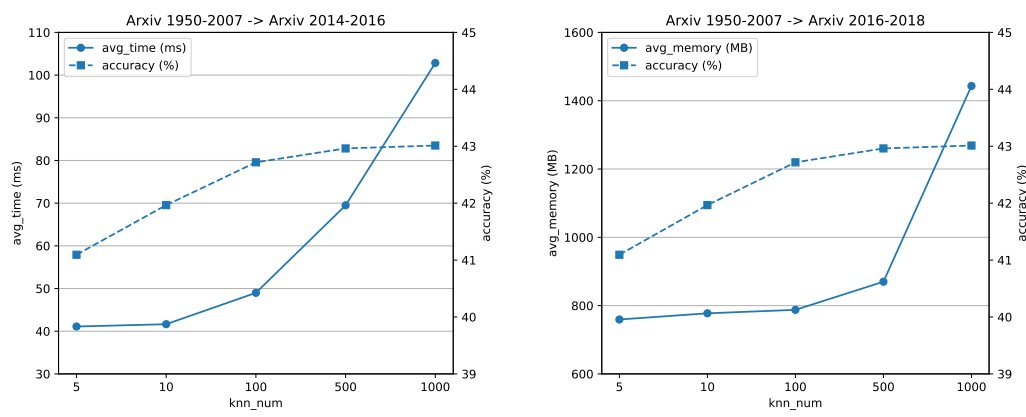

(a) Accuracy and average runtime under different $k$     (b) Accuracy and average memory under different $k$

Figure 8: Accuracy-Efficiency Curve under different numbers of nearest neighbor $k$.

### THE USE OF LARGE LANGUAGE MODELS (LLM)

We commit to using LLMs for text polishing based on prompts. All polished text are double-checked by authors to ensure accuracy, avoid over-claims, and prevent confusion.

