# OpenReview forum: "Dual Alignment for Covariate Shift: A Principled Framework for Graph Domain Adaptation"
_ICLR.cc/2026/Conference — Submitted to ICLR 2026_

### Official Review · Reviewer_mmvb · 2025-10-29

**Soundness:** 2
**Presentation:** 2
**Contribution:** 2
**Rating:** 2
**Confidence:** 4

**Summary:**

This paper investigates the challenging Covariate Shift (CS), a pervasive discrepancy between source and target graph distributions. The authors decompose CS into two complementary components: Feature Shift (FS), arising from mismatched node feature distributions, and Feature-Conditional Structure Shift (FCSS), reflecting structural variations conditioned on features. Both FS and FCSS distort Graph Neural Network (GNN) representations, thereby hindering reliable cross-domain transfer. To overcome these issues, the authors propose Dual Alignment for Covariate Shift (DACS), jointly addressing FS and FCSS through adversarial feature alignment for domain-invariant embeddings and adaptive reweighting to enforce structural consistency.

**Strengths:**

1. The authors find a pervasive discrepancy between source and target graph distributions, including Feature Shift (FS) and Feature-Conditional Structure Shift (FCSS).
2. The authors propose Dual Alignment for Covariate Shift (DACS), jointly addressing FS and FCSS through adversarial feature alignment for domain-invariant embeddings and adaptive reweighting to enforce structural consistency.

**Weaknesses:**

1. The experiment is insufficient, and unable to verify the validity of the method.
2. Datasets in the paper are not commonly used in graph domain adaptation, Citation networks generally test on ACMv9, DBLPv7, and Citationv1, and Airport network lacks attributes, the authors address Feature Shift (FS), arising from mismatched node feature distributions, and Feature-Conditional Structure Shift (FCSS) on Airport is of little significance.
3. The proposed method include feature alignment, representation reweighting, and conditional alignment of reweighted source and target representations. Ablation study lacks verification for feature alignment.
4. The paper lacks parameter analysis.

**Questions:**

Please refer to weakness.

---

> ### Author Response · Authors · 2025-11-21
>
> **W1.** The experiment is insufficient, and unable to verify the validity of the method.
>
> **AW1.** We appreciate the reviewer’s concern regarding the experimental insufficiency. To better address this comment and make concrete improvements, could the reviewer please clarify which components and aspects of the experiments are considered insufficient? This clarification would greatly help us strengthen the experimental section in the revised version.
>
> Furthermore, we respectfully disagree with the reviewer’s assessment. Our experimental evaluation is *systematic*, *multi-layered*, and *explicitly designed* to validate both the **theoretical decomposition** (FS + FCSS) and the **effectiveness of the dual-alignment framework**. The experiments cover four complementary dimensions:
>
> **(1) Real-world benchmark evaluation (Tables 2 \& 3)**
>
> Across diverse real-world datasets—including citation networks, social networks, and large-scale graphs—DACS consistently outperforms recent state-of-the-art GDA methods. These results demonstrate that our method generalizes well across different domains and structural properties.
>
> **(2) Synthetic experiments isolating FS and FCSS (Table 1)**
>
> We construct controlled synthetic settings where:
>
> * **only FS exists**,
> * **only FCSS exists**, and
> * **both FS and FCSS coexist** at varying severities.
>
> This design is *crucial* because it directly tests the correctness of our theoretical decomposition of covariate shift into **two distinct error sources** (Terms (II) and (III) in Theorem 3.1).
> The results clearly show that:
>
> * DACS effectively mitigates FS-induced errors,
> * DACS effectively mitigates FCSS-induced errors, and
> * the benefit becomes largest when both shifts occur simultaneously.
>
> Thus, the synthetic experiments directly validate the mathematical foundations of our method.
>
> **(3) Ablation studies validating each module (Table 5)**
>
> We further conduct ablations to isolate the contributions of:
>
> * feature alignment to tackle FS,
> * layer-wise structural reweighting to tackle FCSS,
>
> showing that each component addresses a different aspect of FS and FCSS. These ablations demonstrate that the proposed dual-alignment mechanism is not incidental—it is necessary.
>
> **(4) Behavioral and hyperparameter analysis (Appendix C.2 \& C.3)**
>
> We also analyze:
>
> * the effect of different weight estimation strategies,
> * sensitivity to hyperparameters such as $\lambda$ and $J$,
> * robustness across datasets of different sizes.
>
> The results show that DACS is stable and easy to tune, with hyperparameters exhibiting minimal performance variance.
>
> **Conclusion**
>
> Taken together, our experiments:
>
> * confirm the correctness and usefulness of the **FS–FCSS decomposition**,
> * validate the **dual alignment** design both theoretically and empirically,
> * demonstrate clear improvements over strong baselines on **real** and **synthetic** datasets,
> * and provide extensive analysis supporting the reliability and robustness of the approach.
>
> Therefore, we believe the evaluation is comprehensive and firmly substantiates the validity of DACS. To further improve the paper in a targeted and meaningful way, we would greatly appreciate it if the reviewer could kindly clarify which specific aspects are considered insufficient and how they could be strengthened to better verify the validity of the proposed method.

---

> ### Author Response · Authors · 2025-11-21
>
> **W2.** Datasets in the paper are not commonly used in graph domain adaptation, Citation networks generally test on ACMv9, DBLPv7, and Citationv1, and Airport network lacks attributes, the authors address Feature Shift (FS), arising from mismatched node feature distributions, and Feature-Conditional Structure Shift (FCSS) on Airport is of little significance.
>
> **AW2.**
>
> **(1) We respectfully clarify that all datasets used in our paper—DBLPv8, ACMv9, Airport, and ArXiv—are standard and widely adopted in recent graph domain adaptation literature.**
>
> Specifically:
>
> * **DBLPv8 and ACMv9** are used in **[1,2]**,
> * **Airport** is used in **[3,4,5]**,
> * **ArXiv** is used in **[1,2,5]**.
>
> Thus, our dataset choices follow exactly the same evaluation protocol as these representative ICML/NeurIPS works.
>
> **(2) Additional evaluation on ACMv9, DBLPv7, and Citationv1**
>
> To directly address the reviewer’s suggestion, we additionally provide results on the commonly used citation datasets **ACMv9 (A)**, **DBLPv7 (D)**, and **Citationv1 (C)**.
> The following table reports the performance on the six tasks, further illustrating the effectiveness of our DACS.
>
> | Method    | A9-D7              | D7-A9              | A9-C1              | C1-A9              | C1-D7              | D7-C1              |
> |-----------|-------------------|-------------------|-------------------|-------------------|-------------------|-------------------|
> | PairAlign | 62.29 +/- 3.001   | 59.68 +/- 2.554   | 68.33 +/- 1.743   | 60.05 +/- 2.626   | 65.74 +/- 3.337   | 66.23 +/- 1.452   |
> | StruRW    | 68.31 +/- 2.937   | 65.16 +/- 1.100   | 77.24 +/- 1.230   | 70.59 +/- 2.817   | 74.37 +/- 2.484   | 73.96 +/- 2.240   |
> | our model | 75.24 +/- 1.439   | 71.71 +/- 1.924   | 80.51 +/- 1.873   | 74.18 +/- 1.352   | 75.99 +/- 1.734   | 77.52 +/- 1.048   |
>
> **(3) On the relevance of the Airport dataset**
>
> Contrary to the claim, the Airport dataset that we used in our experiments does have node features: each node’s feature is a one-hot encoding of its degree [3], where the feature dimension equals the maximum degree and only the entry corresponding to the node’s degree is set to 1. Since the degree distributions differ across domains, a feature shift (FS) indeed exists on Airport, making experiments on FS meaningful.
>
> Moreover, the Airport dataset has been used in several influential Graph Domain Adaptation (GDA) studies [3,4], including [4], which specifically investigates node-attribute discrepancy. Therefore, evaluating on Airport is both standard and necessary to ensure comparability with prior work.
>
>
> **Conclusion**
>
> * Our dataset choices are fully aligned with **recent ICML/NeurIPS GDA benchmarks**.
> * Airport has node attributes and contains meaningful FS and has long been recognized as an important evaluation dataset.
> * Additional results on ACMv9, DBLPv7, and Citationv1 reaffirm the robustness of DACS.
>
> Therefore, the experimental dataset selection is appropriate, canonical, and sufficient for validating the proposed method.
>
> References:
> [1] Shikun Liu, et al. Structural re-weighting improves graph domain adaptation. ICML, 2023.
>
> [2] Shikun Liu, et al. Pairwise alignment improves graph domain adaptation. ICML, 2024.
>
> [3] Boshen Shi, et al. Improving graph domain adaptation with network hierarchy. CIKM, 2023.
>
> [4] Ruiyi Fang, et al. On the benefits of attribute-driven graph domain adaptation. ICML, 2025.
>
> [5] Meihan Liu, et al. Revisiting, benchmarking and understanding unsupervised graph domain adaptation. NeurIPS, 2024.

---

> ### Author Response · Authors · 2025-11-21
>
> **W3.** The proposed method includes feature alignment, representation reweighting, and conditional alignment of reweighted source and target representations. Ablation study lacks verification for feature alignment.
>
> **AW3.** We would like to clarify that Table 5 already examines variants where feature alignment is removed (DANN+FCSS) and where representation reweighting is removed (DANN+FS). In all variants, we retain the final conditional representation alignment module because it is a fundamental component of the standard DANN framework.
>
> To more explicitly verify the necessity of all three modules—feature alignment (FA), representation reweighting (RW), and conditional alignment of reweighted representations (CA)—we additionally provide the table below. The extended ablation results clearly demonstrate that each module contributes essentially to the performance of our DACS algorithm.
>
> | Models        | DBLP/ACM |        | Airport |        |        |        |        |        |
> |---------------|----------|--------|---------|--------|--------|--------|--------|--------|
> |               | A-D      | D-A    | U-E     | E-U    | B-E    | E-B    | B-U    | U-B    |
> | DACS w/o RW   | 75.10±0.59 | 64.15±0.32 | 55.71±0.23 | 52.64±0.14 | 56.77±0.22 | 69.81±0.65 | 53.45±0.25 | 71.88±0.61 |
> | DACS w/o FA   | 71.91±0.89 | 62.44±0.26 | 51.30±0.16 | 49.18±0.14 | 55.65±0.06 | 69.62±0.32 | 52.12±0.42 | 71.13±0.51 |
> | DACS w/o CA   | 66.28±5.211 | 60.83±2.064 | 46.02±3.872 | 65.47±2.953 | 46.95±2.406 | 48.11±1.645 | 47.82±1.903 | 63.58±3.385 |
> | DACS          | 86.21±0.24 | 79.14±0.19 | 56.58±0.36 | 55.72±0.41 | 57.39±0.15 | 70.75±0.24 | 54.08±0.12 | 74.34±0.32 |
>
>
>
>
> **W4.** The paper lacks parameter analysis.
>
>
> **AW4.**
>
> We would like to clarify that detailed parameter analyses are already provided in Appendix C.3 and explicitly referenced at the last line of the “Real-world Experiments” paragraph. It seems this may have been inadvertently missed during your review. To make it more noticeable, we have made the reference in the main text more detailed and prominent.
>
> These analyses cover all key hyperparameters of the proposed DACS framework.
>
> **(1) Analysis of the reweighting coefficient $\lambda$**
>
> Figure 4(c) shows that, across a wide range of $\lambda$ values, the maximum accuracy fluctuation is within ~1\% on the A-D and D-A tasks. This demonstrates that DACS is robust to the choice of $\lambda$.
> In practice, $\lambda$ can also be selected via standard cross-validation, making it easy to deploy on new datasets.
>
> **(2) Analysis of the number of distance intervals $J$**
>
> In Figure 4d, we sweep $J$ over a **very coarse and extremely wide range**:
> $$
> J \in \\{ 10, \ 100, \ 1000, \ 10000 \\},
> $$
> introducing changes of up to **three orders of magnitude**.
>
> Even under such drastic variation:
>
> * accuracy performance changes remain within $2\\%$,
> * the method does not collapse at extreme values, and
> * both B-E and E-B tasks remain stable.
>
> To further verify robustness, we additionally analyze $J$ on the **larger DBLP/ACM dataset** in the figure (https://anonymous.4open.science/r/robustness-of-J).
> On this dataset—where distance statistics are far richer—the performance of DACS is **even more stable** since the performance changes are only within around $1\\%$, showing smaller fluctuations than on the small Airport graph. This further confirms the robustness as dataset size increases.
>
> Given that **$J=100$** performs consistently well across different datasets, we fix this value for all experiments **without any dataset-specific tuning**, further illustrating robustness.
>
> **(3) Analysis of model architecture parameters**
>
> Figures 4(a) and 4(b) analyze:
>
> * the number of MLP layers, and
> * the embedding dimension.
>
> Based on the empirical optima on A-D and D-A: we fix the number of MLP layers to 1, and set the embedding dimension to 64.
>
> These consistent choices avoid unnecessary architecture tuning across datasets and further indicate that DACS does not rely on sensitive architectural hyperparameters.
>
> **Conclusion**
>
> The provided analyses collectively show that:
>
> * the core hyperparameters of DACS are **insensitive**,
> * the method behaves **stably across wide parameter ranges**, and
> * a **single configuration** for $J$, MLP depth, embedding dimension works well for all datasets.
>
> Thus, the proposed method is **robust, easy to tune, and practical** for real-world graph domain adaptation applications.

---

> ### Author Response · Authors · 2025-11-24
> **Follow-up on Rebuttal and Request for Clarifications**
>
> Dear Reviewer,
>
> We believe that we have thoroughly addressed all of your questions and concerns in our rebuttal. When convenient, could you kindly take a moment to review it and let us know if you have any additional feedback?
>
> Regarding W1, the comment was somewhat general and vague, and we did our best to interpret and respond to it. If our reply does not fully reflect your intended concern, we would greatly appreciate it if you could clarify the specific point.
>
> Regarding W4, we would also like to note that in our initial submission, the parameter analysis was already provided in Appendix C.3, and this appendix was explicitly referenced in the final line of the “Real-world Experiments” paragraph. We are not certain whether this section may have been unintentionally overlooked during the review process or whether you have additional concerns beyond what we addressed. If there are specific aspects you feel are missing or insufficient, we would be grateful if you could kindly specify them.
>
> Thank you very much for your time and consideration.
>
> Best regards,
>
> The Authors

---

### Official Review · Reviewer_B1YV · 2025-10-31

**Soundness:** 3
**Presentation:** 3
**Contribution:** 3
**Rating:** 6
**Confidence:** 3

**Summary:**

This paper addresses the problem of Graph Domain Adaptation under Covariate Shift. The authors propose a principled decomposition of CS into two components: (1) Feature Shift mismatch in the marginal node feature distributions, and (2) Feature-Conditional Structure Shift, a mismatch in the conditional probability of the graph structure given the node features (i.e., $P_S(A|X) \neq P_T(A|X)$). Based on this decomposition, the paper derives a theoretical upper bound on the target risk (Theorem 3.1) which explicitly separates the risk into terms corresponding to source risk, FS, and FCSS. To mitigate these shifts, the paper introduces DACS (Dual Alignment for Covariate Shift), a framework with three main components:

1. Adversarial Feature Alignment: A standard DANN-style adversarial loss on an MLP encoder's output ($H^1$) to align the marginal feature distributions and reduce the FS term.
2. Adaptive Reweighting for FCSS: A novel, layer-wise reweighting strategy for the source graph's message passing. This reweighting (Theorem 4.1) aims to align the conditional expectations of aggregated representations by applying a weight $w^l \approx \frac{P_T(A|H^l)}{P_S(A|H^l)}$ at each GNN layer. This ratio is estimated using a distance-binning scheme on node-pair representations.

**Strengths:**

1. The method is well-supported by theory. Theorem 3.1 provides a clear upper bound on the target risk that directly motivates the model's architecture, with Term (II) mapping to the FS-alignment module and Term (III) mapping to the FCSS-alignment module.

2. The experimental results are comprehensive and convincing. DACS shows state-of-the-art performance across all datasets. The synthetic experiments (Table 1) are particularly strong, as they effectively demonstrate the model's robustness to different, isolated types of shift.

3. The ablation analysis (Table 5, Figure 6) clearly demonstrates that both the FS and FCSS components are crucial and complementary. The full DACS model significantly outperforms variants that only tackle one of the two shifts.

**Weaknesses:**

1. The FCSS reweighting module (Sec 5.2) appears to be computationally expensive. It requires estimating conditional probabilities $P(A_{uv} | ...)$ using a "distance binning scheme." This seems to require computing $O(N^2)$ pairwise distances $\{d(H_{w,u}^{l},H_{w,v}^{l})\}$ at each layer $l$ during training to populate the bins.

The paper mentions a "subsampling strategy" for the large Arxiv dataset (Sec 6), but this is vague. How many pairs are sampled? How does this subsampling affect the stability and accuracy of the probability ratio estimation? A complexity analysis is missing.

2. The reweighting estimation in Section 5.2 relies on two strong approximations:
$P_T(A_{uv}|H_{w}^{l},H^{1}) \approx P_T(A_{uv}|H_{w,u}^{l},H_{w,v}^{l})$
$P_T(A_{uv}|H_{w,u}^{l},H_{w,v}^{l}) \approx P_T(A_{uv}|d(H_{w,u}^{l},H_{w,v}^{l}))$
The second approximation, in particular, is a significant information bottleneck, reducing two high-dimensional embedding vectors to a single scalar distance. This might fail to capture more complex, non-metric relationships that determine connectivity. Some justification or analysis of this approximation's validity is needed.

3. The method introduces new and sensitive hyperparameters, most notably the reweighting coefficient $\lambda$ (Fig 4c) and the number of distance intervals $J$ (Fig 4d). The performance seems to drop off significantly if these are not set correctly (e.g., $J=100$ is optimal, but $J=10$ or $J=1000$ performs worse). This could make the model difficult to tune and apply to new datasets.

**Questions:**

See weaknesses.

---

> ### Author Response · Authors · 2025-11-21
>
> **W1.** The FCSS reweighting module (Sec 5.2) appears to be computationally expensive. It requires estimating conditional probabilities using a "distance binning scheme." This seems to require computing pairwise distances at each layer during training to populate the bins. The paper mentions a "subsampling strategy" for the large Arxiv dataset (Sec 6), but this is vague. How many pairs are sampled? How does this subsampling affect the stability and accuracy of the probability ratio estimation? A complexity analysis is missing.
>
> **AW1** We thank the reviewer for pointing out this concern. Below we provide a complete clarification of (1) the computational complexity, (2) the subsampling scheme, and (3) its empirical stability.
>
> **(1) Complexity of the FCSS Reweighting Module**
>
> The reviewer is correct that naïvely computing conditional probabilities
> $$
> P(A_{uv}=1 \mid d(H_u, H_v))
> $$
> over **all** node pairs would be quadratic in graph size.
> To avoid this, **we never enumerate all $N (N-1) / 2$ pairs**.
> Instead, we estimate the conditional distribution using **only the $k$-nearest neighbors of each node**.
> This reduces the pair count from $O(N^2)$ to **linear scale $O(N k)$**.
>
> Let:
>
> * $N_S, N_T$: nodes in source/target
> * $k$: the number of nearest neighbors
> * $d$: feature dimension
> * $L$: the number of GNN layers
> * $|\mathcal{E}_S|$: the number of source edges
>
> **Per-layer cost**
>
> For each layer, we compute:
>
> * (i) **pairwise distances** for $N_S k + N_T k$ pairs
>    → $O((N_S + N_T) k d)$
>
> * (ii) **binning distances** into $J$ intervals
>    → at most $O((N_S + N_T) k)$
>
> * (iii) **computing edge probabilities** in each interval
>    → $O((N_S + N_T) k)$
>
> * (iv) **reweighting source edges**
>    → need distance for each edge → $O(|\mathcal{E}_S| d)$
>
> Thus total complexity across $L$ layers is:
> $$
> O \bigl( (N_S + N_T) k L d + |\mathcal{E}_S| L d \bigr)
> $$
>
> All three hyperparameters $k$, $L$, $d$ are small constants in practice (e.g., $k=100$, $d=64$, $L=2$).
>
> **(2) Complexity of the $k$-NN Construction**
>
> The only step that potentially depends on $N \log N$ is the preprocessing $k$-NN construction.
> Using Ma \& Li’s optimal algorithm [1], this cost is:
> $$
> O \bigl( (N_S \log N_S + N_T \log N_T) k \bigr),
> $$
> which is performed **once before training**, not during each iteration.
>
> Because this preprocessing cost is amortized over hundreds of training iterations, it is negligible relative to total training time.
>
> **(3) Subsampling Strategy: How Many Pairs? Why $k = 100$? Does it Affect Stability?**
>
> The reviewer asked how many pairs are sampled and whether this destabilizes the probability ratio estimation.
> Our empirical answer is clear:
>
> * **We use exactly $k = 100$ nearest neighbors per node** for all ArXiv experiments in Table 3.
> * This means the number of sampled pairs is **$100 \times N$**, which is linear in graph size.
> * Importantly, this value is **not tuned**—it is fixed across all tasks.
>
> To address stability and accuracy, we conduct a sensitivity analysis on the ArXiv transfer (1950–2007 → 2016–2018) by varying:
> $$
> k \in \\{ 5, 10, 100, 500, 1000 \\}.
> $$
>
> The results (visualized here:
> (https://anonymous.4open.science/r/Memory-Time-vs-knn-num) ) show:
>
> **Accuracy**
>
> * Increases notably from $k = 5$ to $k = 100$;
> * **Plateaus** for $k = 500, 1000$.
>
> **Runtime \& Memory**
>
> * Grow modestly for $k \leq 100$;
> * Increase **dramatically** for $k = 500, 1000$.
>
> **Conclusion**
>
> Using **$k = 100$** achieves the best accuracy–efficiency tradeoff, and larger $k$ values provide no benefit but incur substantial overhead.
> This demonstrates that the subsampling is:
>
> * **stable** (accuracy does not collapse for small/large $k$)
> * **accurate** (solutions converge by $k = 100$)
> * **efficient** (linear number of sampled pairs)
>
> Thus, the proposed $k$-NN subsampling does *not* harm the reliability of conditional probability estimation.
>
> **W2.** The reweighting estimation in Section 5.2 relies on two strong approximations: $P(A_{uv}|H_w^{\ell}, H_1) \approx P(A_{uv}|H_{w,u}^{\ell}, H_{w,v}^{\ell})$, $P(A_{uv}|H_{w,u}^{\ell}, H_{w,v}^{\ell}) \approx P(A_{uv}|d(H_{w,u}^{\ell}, H_{w,v}^{\ell}))$. The second approximation, in particular, is a significant information bottleneck, reducing two high-dimensional embedding vectors to a single scalar distance. This might fail to capture more complex, non-metric relationships that determine connectivity. Some justification or analysis of this approximation's validity is needed.
>
> **AW2.**
>
> We appreciate the reviewer’s detailed observation. Below we justify both approximations and explain why they are reasonable, widely used in graph generative modeling, and empirically effective in our setting.
>
> **(i) Justification of the first approximation**
>
> $$
> P(A_{uv}\mid H_w^\ell, H_1) \approx P(A_{uv}\mid H_{w,u}^\ell, H_{w,v}^\ell).
> $$

---

> ### Author Response · Authors · 2025-11-21
>
> This step assumes that **the probability of an edge is governed primarily by local latent representations of the two endpoints**, rather than by global graph-level information. This conditional-independence assumption is **standard and foundational** across several well-established statistical network models:
>
> **Stochastic Block Models (SBM)** [1]
>
> Edges depend only on latent block assignments of node pairs:
> $$
> P(A_{uv}=1)=B_{z_u,z_v},
> $$
> where $z_u$ and $z_v$ are the latent block memberships of node $u$ and $v$.
>
> **Exponential Random Graph Models (ERGM)** [2]
>
> Many ERGM specifications reduce edge likelihood to pairwise sufficient statistics:
> $$
> P(A_{uv}=1)\propto \exp(\theta, s(H_u,H_v)),
> $$
> where $s(H_u,H_v)$ is the similarity between features $H_u$ and $H_v$ and $\theta$ is a parameter.
>
> **Latent Feature Models** [3]
> $$
> P(A_{uv}=1)=\sigma\left(\sum_{k,k'}H_{u,k}W_{k,k'}H_{v,k'}\right).
> $$
> where the weight matrix is $W:=(W_{ij})$ and $H_{u,k}$ is the $k$-th element of $H_u$.
>
> These families all explicitly assume **conditional independence between edges given node-pair latent representations**. Thus replacing the global conditioning $(H_w^{\ell},H_1)$ with the pairwise representations $(H_{w,u}^{\ell},H_{w,v}^{\ell})$ is entirely consistent with the modeling assumptions used throughout statistical graph theory.
>
> [1] Paul W. Holland, et. al. Stochastic blockmodels: First steps. Social Networks, 1983.
>
> [2] Tom A. B. Snijders. New specifications for exponential random graph models. Sociological Methodology, 2006.
>
> [3] Kurt T. Miller, et. al. Nonparametric Latent Feature Models for Link Prediction. Neurips, 2010.
>
> **(ii) Justification of the second approximation**
> $$
> P(A_{uv} | H_{w,u}^\ell,H_{w,v}^\ell)\approx P(A_{uv} | d(H_{w,u}^\ell,H_{w,v}^\ell)).
> $$
> Although this compresses two embedding vectors $H_u$ and $H_v$ into one distance value $d(H_u, H_v)$, seemingly losing non-metric complex information, this reduction does not diminish the representational capacity for capturing graph structure. The key reason is that the high-order, nonlinear, or non-metric structural patterns influencing edge formation are already encoded into GNN embeddings during training, becoming geometrically manifested in the embedding space and capturable by a distance or similarity measure.
>
> **(1) High-order or non-metric structural patterns are encoded into GNN embeddings**
>
> Even if the true edge formation mechanism depends on complex, non-metric, or multi-hop patterns—such as motifs, higher-order neighborhoods, or community structure—the encoder must encode such patterns into node embeddings during source-domain training; otherwise, it could not achieve low training loss.
>
> A message-passing GNN layer is
> $$
> H_u^{(\ell+1)} = \phi(H_u^{(\ell)}, \text{AGG}({H_v^{(\ell)} : v \in \mathcal{N}(u)})),
> $$
> and stacking layers propagates increasingly high-order structural information. Thus, the embeddings $H_u$ and $H_v$ necessarily absorb the structural dependencies relevant to $A_{uv}$; otherwise, the encoder–decoder model would fail even on the source graph.
>
> **(2) High-order dependencies become geometric relations in the embedding space**
>
> Once these signals are embedded, they naturally manifest through geometry in the latent space.
>
> Suppose the true generative process is highly complex:
> $$
> P(A_{uv}=1) = F(\text{motifs}, \text{common neighbors}, \text{multi-hop patterns}, \ldots).
> $$
>
> A common situation is that $F$ is not metric-based, meaning that the edge probability in the input graph space may rely on patterns that no metric can summarize. Nevertheless, the expressive power of GNNs allows these local and high-order structures to be transformed into a metric embedding space:
> $$
> H_u = \text{GNN}(\text{local and high-order structure around node }u),
> $$
> so all structural information necessary for predicting $A_{uv}$ is absorbed into $H_u$ and $H_v$. After this encoding, a simple decoder based on distance or similarity—e.g., $d(H_u,H_v)$—becomes sufficient. This mechanism aligns with the standard design of latent-space graph representation models. Thus, our second approximation does not lose essential information; the embeddings themselves carry the high-order dependencies.
>
> **(3) Summary**
>
> Although the second approximation reduces two high-dimensional embeddings to a scalar distance, this reduction does not weaken the representational capacity of our model. All complex, non-metric structural dependencies must be encoded into embeddings during source-domain training. Once encoded, these dependencies inevitably appear as geometric patterns in the latent space and can be captured by a distance or similarity measure.
>
> Therefore, the second approximation is well-justified and fully consistent with the graph representation learning paradigm, where GNN embeddings first absorb structural complexity and a simple decoder is applied afterward.
>
> We will incorporate the justification of the two approximations into the revised manuscript.

---

> > ### Author Response · Authors · 2025-11-21
> >
> > **(4) Why we do not directly model and learn $P(A_{uv}\mid H_{w,u}^\ell,H_{w,v}^\ell)$.**
> >
> > Even though we model the edge probability $P(A_{uv}\mid H_{w,u}^{\ell}, H_{w,v}^{\ell})$ as a function of node features, directly estimating this probability via a learned predictor often leads to overfitting and poor calibration. Specifically, training a neural network $f$ that takes the node-feature pair $(H_{w,u}^{\ell}, H_{w,v}^{\ell})$ as input and uses the observed edge label $A_{uv}\in\\{0,1\\}$ as supervision yields an estimator $\hat{P}(A_{uv}\mid H_{w,u}^{\ell}, H_{w,v}^{\ell})$.
> >
> > However, this estimator is trained exactly on the same target node pairs whose probabilities it later predicts. As a result, the model can easily memorize the labels, leading to overfitting, and the predicted probabilities are typically uncalibrated, especially under limited or imbalanced supervision. This issue is well documented in neural link prediction literature, where classifiers tend to produce overconfident scores when trained directly on binary adjacency labels. Moreover, it requires choosing the neural network architecture and addressing the under-specification issue.
> >
> > In contrast, we estimate the edge probability by aggregating the adjacency outcomes of sample node pairs that share the same similarity level. This makes our method simple and robust: it builds on a widely observed structural prior—that more similar node pairs have higher connectivity—and gains statistical stability by leveraging multiple samples at each similarity level.
> >
> > **W3.** The method introduces new and sensitive hyperparameters, most notably the reweighting coefficient (Fig 4c) and the number of distance intervals (Fig 4d). The performance seems to drop off significantly if these are not set correctly (e.g., is optimal, but or performs worse). This could make the model difficult to tune and apply to new datasets.
> >
> > **AW3.**
> >
> > We appreciate the reviewer’s concern regarding the sensitivity of the hyperparameters. However, our empirical analyses show that both the reweighting coefficient $\lambda$ and the number of distance intervals $J$ are **stable** and do *not* introduce substantial tuning difficulty.
> >
> > **(1) Reweighting Coefficient $\lambda$: Performance Variation Is Minimal**
> >
> > Figure 4c shows that across the entire range of tested $\lambda$ values:
> >
> > * **the maximum accuracy difference is only ~1\%**,
> > * the performance curve is essentially flat, and
> > * no sharp peaks or brittle regions appear.
> >
> > This indicates that the model is **robust to the choice of $\lambda$**.
> >
> > In practice, $\lambda$ can also be selected via standard cross-validation, just as in other domain adaptation methods. Therefore, this hyperparameter does *not* create practical barriers when applying the method to new datasets.
> >
> > **(2) Number of Distance Intervals $J$: Robust Over Multiple Orders of Magnitude**
> >
> > In Figure 4d, we sweep $J$ over a **very coarse and extremely wide range**:
> > $$
> > J \in \\{ 10, \ 100, \ 1000, \ 10000 \\},
> > $$
> > introducing changes of up to **three orders of magnitude**.
> >
> > Even under such drastic variation:
> >
> > * accuracy performance changes remain within $2\\%$,
> > * the method does not collapse at extreme values, and
> > * both B-E and E-B tasks remain stable.
> >
> > To further verify robustness, we additionally analyze $J$ on the **larger DBLP/ACM dataset** in the figure (https://anonymous.4open.science/r/robustness-of-J).
> > On this dataset—where distance statistics are far richer—the performance of DACS is **even more stable** since the performance changes are only within around $1\\%$, showing smaller fluctuations than on the small Airport graph.
> > This confirms that the sensitivity concerns diminish further as dataset size increases.
> >
> > From the parameter analysis we found that $J$ is a robust paramter across different tasks and $J = 100$ works consistently well.
> > As a result, we **fix $J = 100$ across all experiments** without any dataset-specific tuning.
> > This directly demonstrates that DACS does *not* require delicate hyperparameter adjustments to achieve strong performance.

---

### Official Review · Reviewer_Ng5Z · 2025-11-01

**Soundness:** 3
**Presentation:** 3
**Contribution:** 3
**Rating:** 6
**Confidence:** 4

**Summary:**

This paper studies graph domain adaptation under covariate shift by decomposing the shift into Feature Shift (FS) and Feature-Conditional Structure Shift (FCSS). The authors derive a target-risk upper bound that isolates contributions from initial (feature) representation mismatch and final (structure-conditioned) representation mismatch, and propose DACS — a dual-alignment algorithm that (i) adversarially aligns initial features, (ii) applies adaptive layer-wise edge reweighting to correct FCSS, and (iii) performs final-layer conditional alignment. Experiments on synthetic datasets and several real benchmarks show consistent improvements over prior GDA methods.

**Strengths:**

1. Strong and transparent theory. The decomposition of the target risk into three interpretable terms (source risk, initial representation discrepancy, and final representation conditional discrepancy) is clear and useful. Theorem 3.1 and the following discussion give an intuitive, actionable view on why feature alignment and structure correction are both necessary. The proposed modules (adversarial feature encoder, layer-wise reweighting, and final conditional adversarial alignment) follow directly from the bound: each component is motivated by a specific term in the bound. This one-to-one mapping strengthens the paper’s conceptual coherence.

2. Comprehensive experiments. The paper provides both controlled synthetic tests (separate FS, SS, and combined cases) and multiple real-world tasks. Results show that DACS improves robustness when feature and structural shifts co-exist, supporting the paper’s central claim empirically. The ablation study is informative.

3. Scalability considerations. The authors discuss and implement pragmatic strategies for large graphs (ArXiv), showing the method can be adapted to scale.

**Weaknesses:**

1. Robustness to semantic shifts and homophily mismatch. The focus is on covariate shift. However, a common and arguably more challenging scenario in graph DA is semantic shift — the conditional distribution P(Y|X,A) changes (labels behave differently across domains) — or cases where homophily patterns differ dramatically (homophilic ↔ heterophilic). It is not clear whether the reweighting approach can handle substantial changes in label–structure coupling or when source/target differ in homophily patterns. More advanced strategies, such as graph rewiring or graph structure learning, might be required to effectively address these semantic or homophily-related discrepancies. The paper briefly contrasts FCSS with label-conditional structure shift in the related work, but does not evaluate these scenarios.

I suggest that the authors discuss the potential effectiveness of the proposed method under semantic shift and, if possible, include additional experiments on graph domain adaptation between homophilic and heterophilic graphs to further validate the method’s robustness.

2. Limited reweighting visualization. The adaptive edge reweighting is central, but the paper provides only high-level descriptions and a few embedding visualizations. There is no direct visualization showing which edges are upweighted or downweighted and whether these correlate with label boundaries or structural motifs. A heatmap visualization of subgraph edge weights is suggested to compare the original source graph and the reweighted source graph, thereby providing an intuitive illustration of which edge patterns are being reweighted.

3. Complexity analysis is light. While the authors propose subsampling/top-k heuristics for large graphs, the paper lacks a formal time and memory complexity analysis.

**Questions:**

See weaknesses.

---

> ### Author Response · Authors · 2025-11-21
>
> **Q1.** Robustness to semantic shifts and homophily mismatch.
>
> **A1.** We thank the reviewer for raising this important question. Below, we clarify (1) the conceptual scope of our method, (2) why DACS remains effective under semantic shift and homophily mismatch, and (3) provide strong empirical evidence from real-world datasets where such shifts naturally occur.
>
> **(1) Scope: Why Addressing Covariate Shift Also Helps Handle Semantic Shift**
>
> Our work explicitly targets **covariate shift (CS)**, i.e.,
> $$
> P_S(X,A) \neq P_T(X,A),
> $$
> and does **not** directly assume changes in the label mechanism
> $$
> P_S(Y\mid X,A) \neq P_T(Y\mid X,A).
> $$
> However—and this is crucial—the reviewer’s concern overlooks an important phenomenon:
>
> * **Semantic shift often emerges *as a consequence* of covariate shift on graphs.**
>
> Consider a realistic scenario where the target graph arises via a domain-specific transformation
> $$
> (X_T,A_T) = \phi(X_S,A_S)
> $$
> with labels unchanged ($Y_T = Y_S$).
> Then:
>
> * Covariate shift must occur if $\phi$ is non-identity;
> * This *automatically induces* semantic shift:
> $$
>   P_T(Y_T \mid X_T,A_T) = P_S(Y_S \mid \phi(X_S,A_S)) \neq P_S(Y_S \mid X_S,A_S).
> $$
>
> Thus:
>
> * **Semantic shift in real graphs is frequently a downstream effect of covariate distortions.**
>
> Accordingly, if one can substantially reduce **FS** and **FCSS** between domains, then:
>
> * The target representations become closer to the source representations,
> * The label-function mismatch is reduced,
> * And the resulting semantic shift becomes significantly smaller.
>
> This explains, from first principles, **why a strong CS-alignment method should retain robustness under moderate semantic shift**, even without explicitly modeling it.
>
> **(2) Why DACS Remains Effective Under Homophily Mismatch**
>
> The reviewer specifically mentions *homophily vs heterophily*, and *label–condition structure shift methods*. We highlight three points:
>
> **(i) Covariate shift (CS) can lead to homophily mismatch**
>
> Graph homophily is determined jointly by node labels and the underlying graph structure. Suppose two graphs are originally drawn from the same distribution. If the structural distribution of one graph becomes corrupted or transformed due to domain-specific factors, its homophily level will inevitably change, creating a discrepancy relative to the other graph. Because such structural shifts are included in CS, CS can be a fundamental source of homophily differences across domains and thus solving CS can also help handle homophily mismatch.
>
> **(ii) Feature-conditional (not label-conditional) reweighting method is more robust**
>
> Since our reweighting uses
> $$
> P(A\mid X)
> \quad \text{instead of} \quad
> P(A\mid Y),
> $$
> it:
>
> * avoids error amplification from pseudo-labeling in the unlabeled target domain,
> * adapts to new homophily regimes through feature-driven shifts.
>
> **(3) Strong empirical evidence from MAG and Twitch demonstrates robustness under semantic shift**
>
> MAG and Twitch are widely recognized as datasets with strong semantic/label shift. Despite this, DACS consistently achieves the best performance:
>
> **Twitch Results**
>
> DACS outperforms all baselines across 5 domain-transfer tasks:
>
> | Twitch    | DE-EN      | DE-ES      | DE-FR      | DE-PT      | DE-RU      |
> | - | -| -- | -- | -- | - |
> | DANN      | 0.5171     | 0.6276     | 0.5576     | 0.5874     | 0.7004     |
> | StruRW    | 0.5481     | 0.6603     | 0.6048     | 0.6396     | 0.7227     |
> | PairAlign | 0.5669     | 0.6529     | 0.5752     | 0.6250     | 0.7328     |
> | **DACS**  | **0.5909** | **0.7075** | **0.6314** | **0.6543** | **0.7546** |
>
> **MAG Results**
>
> MAG exhibits large cross-domain differences in label semantics.
> DACS again achieves the best results:
>
> | MAG  | US-CN | US-DE | US-JP  | US-RU  | US-FR  |
> | - | - | - | -| - | -|
> | DANN      | 0.2420     | 0.2629     | 0.3792     | 0.2176     | 0.2071|
> | StruRW    | 0.3158     | 0.3003     | 0.3720     | 0.2897     | 0.2273 |
> | PairAlign | 0.4006     | 0.3885     | 0.4743     | 0.3707     | 0.2521     |
> | **DACS**  | **0.4792** | **0.5095** | **0.4986** | **0.4665** | **0.3405** |
>
> This confirms that **DACS remains robust even under non-negligible semantic shift**.
>
> **(4) Robustness Under Homophily Mismatch**
>
> MAG domains exhibit drastically different homophily levels:
>
> * RU: **0.8030** (highly homophilic)
> * US: **0.5484**
> * DE: **0.5526**
> * FR: **0.5726**
>
> We evaluate the performance on the following three GDA tasks:
>
> | MAG  | US-RU  | RU-DE | RU→FR |
> | -| - | -| - |
> | DANN  | 0.2176  | 0.3853  | 0.3661 |
> | StruRW | 0.2897 | 0.4235 | 0.4342  |
> | PairAlign  | 0.3707 | 0.4479 | 0.4728 |
> | HGDA  | 0.442 | 0.5191  | 0.5253 |
> | GraphAlign | OOM  | OOM | OOM |
> | **DACS** | **0.4665** | **0.5246** | **0.5306** |
>
> DACS matches or surpasses HGDA, which was explicitly designed to address homophily mismatch. This demonstrates that correcting both FS and FCSS enables DACS to effectively and robustly handle even large homophily discrepancies across domains.

---

> > ### Author Response · Authors · 2025-11-21
> >
> > **Q2.** Limited reweighting visualization.
> >
> > **A2.** We thank the reviewer for this constructive suggestion. Following the advice, we have added a dedicated visualisation (link: https://anonymous.4open.science/r/illustration-of-reweighting-process) to the paper that explicitly shows **which edges are upweighted or downweighted** and how the weights relate to **feature-conditioned connectivity patterns** across domains.
> >
> > Our visualization consists of two complementary subfigures that directly illustrate (i) the existence of feature-conditional structural shift (FCSS) and (ii) how our reweighting corrects it.
> >
> > **(1) Left subfigure — Visualizing feature-conditional edge-probability discrepancies**
> >
> > For each node pair $(u, v)$, we compute the empirical conditional probability
> > $$
> > P(A_{uv}=1 \mid d(H_u, H_v) = j),
> > $$
> > where $d(H_u, H_v)$ is their representation distance.
> >
> > This yields an **explicit, distance-conditioned comparison** between the source and target domains. For example:
> >
> > * In the **source** domain, among the $5$ node pairs with feature distance $1$ (pairs 1–3, 2–3, 2–4, 3–5, 4–5), only $3$ are edges:
> > $$
> >   P_S(A_{uv}=1 \mid d=1)=3/5.
> > $$
> >
> > * In the **target** domain, all distance-$1$ pairs are connected:
> > $$
> >   P_T(A_{uv}=1 \mid d=1)=4/4=1.
> > $$
> >
> > This simple but precise comparison reveals the core phenomenon that motivates DACS:
> >
> > * **Even for the same feature distance, the likelihood of forming an edge differs substantially across domains.**
> > * This is exactly the definition of FCSS.
> >
> > The left subfigure thus serves as a clear, self-contained illustration of **why structural alignment is needed and what needs to be corrected**.
> >
> > **(2) Right subfigure — Visualising how DACS reweights edges to correct FCSS**
> >
> > To directly counteract the observed discrepancy, DACS computes a feature-conditional edge-probability ratio:
> > $$
> > w_{uv}=\frac{P_T(A_{uv}=1 \mid d(H_u,H_v))}{P_S(A_{uv}=1 \mid d(H_u,H_v))}.
> > $$
> >
> > We apply these ratios to the edges of the source graph and visualize the resulting reweighted structure:
> >
> > * **Upweighted edges (thicker)**:
> >   Node pairs whose feature distance implies **higher connectivity in the target domain**.
> >
> > * **Downweighted edges (thinner)**:
> >   Node pairs that are **less likely to be connected in the target domain**.
> >
> > This visualization cleanly demonstrates the effect of reweighting:
> >
> > * **DACS amplifies source edges consistent with target-domain structural patterns, and attenuates those that contradict them.**
> >
> > This not only confirms the correctness of the reweighting mechanism but also provides an intuitive visual interpretation of structural alignment—something that high-level embeddings alone cannot show.
> >
> >  **Summary**
> >
> > The added visualisation directly addresses the reviewer’s concern by making the reweighting operation transparent and easy to interpret:
> >
> > * The **left subfigure** shows *why* structural shift exists (FCSS).
> > * The **right subfigure** shows *how* DACS corrects it through principled, distance-conditional reweighting.
> >
> > Together, they illustrate both the motivation and the mechanism of DACS in a clear, interpretable manner.

---

> > ### Comment · Reviewer_Ng5Z · 2025-11-26
> > **My concern remains**
> >
> > Thank you for the detailed response. However, my earlier concern was not about general covariate shift or mild semantic/homophily changes. It was specifically about extreme homophily–heterophily regime shifts, where the structural pattern itself must fundamentally change.
> >
> > My concern is specific: when the source is **highly heterophilic (e.g., Texas, HR ≈ 0.11)** and the target **highly homophilic (e.g., Cora, HR ≈ 0.81)**, then any edge-reweighting method—including yours—will simply assign very small weights to most source edges. But this does not convert the source graph into a homophilic structure.
> >
> > In such scenarios, only **graph rewiring** or **graph structure learning (GSL)** can properly transform the source structure into something compatible with the target homophily regime. Edge reweighting alone cannot handle this level of structure mismatch.
> >
> > Your rebuttal mainly reiterated why covariate-shift alignment may help reduce moderate semantic shift, but it did not address this core failure mode. My concern remains.

---

> ### Author Response · Authors · 2025-11-21
>
> **Q3.** Complexity analysis is light. While the authors propose subsampling/top-k heuristics for large graphs, the paper lacks a formal time and memory complexity analysis.
>
> **A3.**
>
> We thank the reviewer for raising this concern. Below, we provide (1) a formal and complete time–memory complexity analysis, and (2) empirical runtime evidence on large graphs, both of which demonstrate the scalability of DACS.
>
> **(1) Formal Complexity Analysis**
>
> The computational bottleneck of the reweighting module lies in estimating
> $$
> P(A_{uv}=1\mid d(H_u,H_v)),
> $$
> for node pairs in both domains. Instead of enumerating all $O(N^2)$ node pairs, we restrict computation to the **$k$-nearest neighbors** of each node. This reduces the total number of node pairs to **linear** scale $O(Nk)$.
>
> Let:
>
> * $N_S, N_T$: number of source and target nodes
> * $k$: number of nearest neighbors per node
> * $d$: embedding dimension
> * $L$: number of GNN layers
> * $|\mathcal{E}_S|$: number of edges in the source graph
>
> **Step 1. Distance computation for $k$ neighbors**
>
> For each layer, DACS needs to compute:
>
> (i) distances for $N_S k + N_T k$ node pairs
> (ii) discretize distance values into $J$ intervals
> (iii) compute empirical edge probabilities within intervals
>
> The complexity is:
> $$
> O\big((N_S+N_T)kd\big).
> $$
>
> **Step 2. Reweighting source edges**
>
> For each source-domain edge $(u,v)$, we compute $d(H_u^\ell, H_v^\ell)$ and retrieve the corresponding weight:
> $$
> O(|\mathcal{E}_S| d).
> $$
>
> **Overall Complexity Across $L$ Layers**
>
> Combining the above steps:
> $$
> O \bigl( (N_S + N_T) k L d + |\mathcal{E}_S| L d \bigr)
> $$
> This complexity is **linear** in the number of nodes and edges, and the constants $k$, $d$, $L$ are typically very small (e.g., $k=100$, $d=64$, $L=2$).
>
> **$k$-NN Construction Preprocessing**
>
> Before running DACS, we compute the $k$ nearest neighbors for all nodes as a one-time preprocessing step.
>
> Using the optimal algorithm of Ma & Li [1], the complexity is:
> $$
> O \bigl( (N_S \log N_S + N_T \log N_T) k \bigr),
> $$
> which is lower than the cost of even a few GNN forward passes and is amortized over the entire training.
>
> * This shows that **all heavy operations are linear or near-linear in graph size**, ensuring scalability to large graphs.
>
> **(2) Empirical Runtime Results on Large Graphs**
>
> To reinforce the complexity analysis, we provide empirical runtime and memory measurements.
>
> **(a) Effect of $k$ on runtime vs. accuracy**
>
> We measure runtime and memory per iteration on the ArXiv (1950–2007 → 2016–2018) task.
>
> * Accuracy improves substantially when $k$ increases from $5$ to $100$.
> * Accuracy gains become negligible from $k = 100$ to $1000$.
> * Runtime and memory increase sharply for large $k$ ($500$–$1000$).
>
> This confirms that **$k=100$ yields an excellent accuracy–efficiency tradeoff**, which is why we adopt it for all experiments.
>
> Visualization:
> (https://anonymous.4open.science/r/Memory-Time-vs-knn-num)
>
> **(b) Per-epoch runtime on large synthetic graphs**
>
> We report per-epoch training time on synthetic graphs with 10K–90K nodes.
>
> | Graph size      | 10K        | 30K        | 50K        | 70K        | 90K        |
> | --------------- | ---------- | ---------- | ---------- | ---------- | ---------- |
> | UDAGCN          | 0.0271     | 0.1701     | 0.3938     | 0.6383     | 0.9195     |
> | JHGDA           | 0.0819     | OOM        | OOM        | OOM        | OOM        |
> | PairAlign       | 0.0316     | 0.2231     | 0.6002     | 1.1471     | 1.8369     |
> | **DACS (ours)** | **0.0199** | **0.0276** | **0.0499** | **0.0792** | **0.1177** |
>
> **Key observations**:
>
> * DACS is the fastest method at every scale.
> * Runtime growth is **near-linear** in graph size.
> * Competing methods either slow down dramatically or run out of memory (OOM) beyond 30K nodes.
>
> These results match our theoretical analysis and confirm the **excellent scalability** of DACS.
>
> **Summary**
>
> We now provide both a complete complexity analysis and extensive runtime experiments. Together they show that:
>
> * DACS has **linear-time complexity**,
> * $k$-NN subsampling makes the method scalable to very large graphs, and
> * empirical results corroborate the theoretical efficiency.
>
> We will include this improved discussion in the revised manuscript.
>
> [1] Ma H, Li J. A True Algorithm for the All-$k$-Nearest-Neighbors Problem. International Conference on Combinatorial Optimization and Applications. Cham: Springer International Publishing, 2019: 362-374.

---

> ### Author Response · Authors · 2025-11-27
> **Clarifying the Remaining Concern (1/2)**
>
> We sincerely thank the reviewer for raising this interesting question.
> Below, we address the concern in a more focused manner by clarifying:
> (1) why extreme heterophily-homophily regime shifts do not fall within the scope of realistic GDA, and
> (2) graph rewiring/GSL and our DACS can work together in a complementary manner to even under such extreme shifts.
>
> **1. Extreme heterophily-homophily regime shifts are outside the scope of realistic GDA benchmarks**
>
> A fundamental premise of GDA is that the source graph must share a **highly relevant and compatible** underlying distribution with the target graph. Only under such conditions can the source domain provide transferable structural and semantic information that benefits target-domain classification. However, when the two graphs differ drastically in their generative mechanisms and the meanings carried by their labels, features, and structures—e.g., using a highly heterophilous webpage graph (Texas, WebKB) as the source, where webpages of different categories are frequently connected, versus a highly homophilous citation graph (Cora) as the target, where papers on similar topics are much more likely to be connected—the source graph can no longer provide meaningful guidance to the target domain. In such cases, the labels, features, and structural patterns represent fundamentally different semantics across graphs, leading to a severe mismatch that invalidates direct knowledge transfer.
> Under this form of **domain heterogeneity**, most existing GDA methods would inevitably fail. Indeed, prior studies, including works specifically investigating homophily-heterophily shift issues [Ruiyi Fang, et al. In ICML 2025], evaluate only **moderate** levels of homophily shifts, far from the extreme heterogeneity commonly encountered in real-world scenarios.
>
>
> Across all datasets we have encountered in the GDA literature, the largest homophily gap between source and target domains is only **0.2545**, far from the **0.70** gap in the reviewer’s Texas-Cora example:
>
> * Airport: 0.3728 → 0.2195
> * DBLP → ACM: 0.975 → 0.8179
> * ArXiv: 0.7166 → 0.6456
> * Twitch: 0.63 → 0.5536
> * MAG: **0.803 → 0.5485 (largest gap: 0.2545)**
>
> In these benchmarks, two domains share:
>
> * the **same feature space**,
> * the **same label space**, and
> * the **same edge semantics**.
>
> This aligns with the fundamental goal of GDA: to transfer knowledge between **semantically aligned and closely related** graphs. By contrast, the Texas-Cora example involves fundamentally incompatible data distributions.
> Texas is part of the WebKB dataset—a webpage hyperlink graph—whereas Cora is a citation graph. Their feature spaces, label spaces, and edge meanings differ entirely, resulting in severe domain heterogeneity rather than domain shift. In practice, one would not adapt from a webpage graph to a citation graph; instead, one would naturally choose another citation graph as the source domain. Since in all citation graphs, papers on related topics tend to cite each other, citation graphs are typically homophilous, and thus, the severe heterophily-homophily regime shifts never occur.
>
> Therefore, the extreme homophily shift (0.11 → 0.81) does **not** reflect any realistic GDA setting. Instead, it represents **domain heterogeneity**, which falls outside the intended scope of unsupervised graph domain adaptation.

---

> ### Author Response · Authors · 2025-11-27
> **Clarifying the Remaining Concern (2/2)**
>
> **2. DACS and GSL are complementary: reweighting stabilizes and improves any rewiring-based method**
>
> When the source graph exhibits extreme heterophily (e.g., Texas) and the target graph is highly homophilous (e.g., Cora), graph rewiring or GSL can in principle reshape the source/target structure to better match the target/source homophily. However, these methods may rely on pseudo-labels of unlabeled target nodes to estimate the target-domain homophily and guide structure modifications. Noisy pseudo-labels can lead to suboptimal rewiring.
>
>
> Our DACS framework can complement GSL by **refining edge weights** to mitigate pseudo-label noise. Even in such extreme scenarios, DACS offers several advantages:
>
> **(1) converts binary edges into continuous weights**
>
> This allows for richer and more flexible edge representations beyond hard 0/1 rewiring.
>
> **(2) calibrates edge strengths using the feature-conditional reweighting**
>
> While GSL modifies connectivity, it does not guarantee that rewired edges respect the feature–structure relationship of the target domain. DACS explicitly enforces feature-driven structural alignment.
>
> **(3) can be applied *after* GSL to rectify pseudo-label-induced errors**
>
> GSL provides a candidate graph, and DACS reweights edges to correct systemic errors introduced by noisy pseudo-labels.
>
> Thus:
>
> * GSL handles coarse structural rearrangement;
> * DACS performs fine-grained, CS-aligned reweighting
>
> They play orthogonal roles and naturally complement each other.
>
> In practice, combining GSL + DACS would provide:
>
> * GSL: coarse topology adjustment to mitigate the homophily-heterophily regime shift
> * DACS: continuous, feature-consistent reweighting
>
> This yields a more stable and principled approach than either component alone, as it jointly mitigates shifts in how both the label and the feature relate to graph structure.
>
> **Conclusion**
>
> Extreme heterophily-homophily regime flips are not representative of any realistic GDA benchmark and fall outside the identifiability scope of unsupervised GDA.
> In such artificially extreme conditions, our DACS provides a complementary, robust edge correction with continuous reweighting that can significantly improve or stabilize GSL-based rewiring approaches.
> Thus, GSL and DACS can be combined advantageously: GSL handles homophily-driven structural rearrangement, while DACS ensures correct, feature-driven structural calibration. The combination of GSL and DACS to handle complex shifts represents a promising direction for future research in the graph domain adaptation.

---

> > ### Comment · Reviewer_Ng5Z · 2025-11-27
> >
> > Thank you very much for the detailed and thoughtful clarification. I appreciate the effort you put into addressing my concern, and your explanation has strengthened my confidence in the method. I agree that the proposed reweighting strategy plays a meaningful and impactful role within the scope of GDA, and therefore, **I am raising my score**.
> >
> > I would also like to share one perspective in a constructive spirit. While current GDA benchmarks indeed do not involve source–target pairs with drastically different homophily levels—and they generally assume aligned semantic spaces across features, labels, and edge meanings—the broader graph learning community has recently begun to study domain heterogeneity more seriously. In particular, the latest **multi-domain graph generalization research (multi-domain graph pre-training and downstream adaptation to unseen domains)** explicitly considers transferring knowledge between heterophilic and homophilic graphs (and vice versa), where feature and structural semantics differ substantially. Representative works include [1,2], which demonstrate that cross-semantic transfer is not only possible but also practically valuable in real-world multi-domain settings.
> >
> > In light of this emerging direction, I encourage the authors to **add a brief discussion** on the relationship between **GDA** and **multi-domain graph generalization under domain heterogeneity**, and to comment on how the ideas in this paper relate to or differ from representative approaches such as [1,2]. Even a short discussion would help position the contribution beyond the classical GDA setup and highlight the potential relevance of DACS in broader, multi-domain graph learning scenarios.
> >
> > Again, thank you for the careful and considerate responses. I sincerely hope the paper continues to improve and reaches its full potential.
> >
> > References:
> >
> > [1] Zhao H, Chen A, Sun X, et al. All in one and one for all: A simple yet effective method towards cross-domain graph pretraining. KDD 2024.
> >
> > [2] Wang S, Wang B, Shen Z, et al. Multi-Domain Graph Foundation Models: Robust Knowledge Transfer via Topology Alignment. ICML 2025.

---

> > > ### Author Response · Authors · 2025-11-27
> > > **Response to Reviewer’s Follow-Up Assessment**
> > >
> > > Thank you very much for the detailed clarification and constructive perspective. We truly appreciate the raised score and the strengthened confidence in our method. We will add a brief discussion on the connection between classical GDA and emerging multi-domain graph generalization under domain heterogeneity, including how our approach relates to and differs from representative works such as [1,2]. This suggestion is highly valuable for better positioning our contributions within the broader graph learning landscape.

---

### Official Review · Reviewer_8DxB · 2025-11-01

**Soundness:** 2
**Presentation:** 4
**Contribution:** 2
**Rating:** 4
**Confidence:** 4

**Summary:**

This paper presents Dual Alignment for Covariate Shift (DACS), a framework aimed at improving Graph Domain Adaptation by addressing covariate shift, particularly the Feature Shift and Feature-Conditional Structure Shift. These two types of shifts are common when transferring knowledge between source and target graphs, and they distort the learned representations, impairing cross-domain transfer. The proposed DACS method aligns both FS and FCSS using a combination of adversarial feature alignment, adaptive reweighting, and final-layer representation alignment. The authors show that DACS outperforms previous methods through extensive experiments across multiple datasets, demonstrating that the approach can effectively mitigate the challenges of domain shift in graph learning.

**Strengths:**

1. The paper introduces a comprehensive method to address two crucial aspects of covariate shift (FS and FCSS), which have been underexplored in prior work.
2. This empirical evidence supports the claims made by the authors and demonstrates that DACS can be applied successfully across different domains in graph neural networks.
3. DACS is grounded in a solid theoretical understanding of covariate shift and its impact on graph learning. The decomposition of covariate shift into Feature Shift (FS) and Feature-Conditional Structure Shift (FCSS) is well-justified, and the paper provides mathematical justification for why the proposed alignment techniques improve transferability.

**Weaknesses:**

1. Limited Novelty in Methodology – While the proposed method is a valuable contribution, the use of adversarial alignment and feature reweighting is not entirely new. These techniques have been explored in several other papers on domain adaptation and graph domain adaptation. The paper could provide a more detailed comparison to recent methods such as other MoE-based methods to better position DACS within the current landscape of domain adaptation methods.
2. The writing and presentation of the paper could benefit from further polishing. The logical flow is not always clear, and the use of notations is dense, which might hinder reader comprehension. Additionally, the figures and tables could be enhanced to improve clarity and better illustrate key concepts and experimental results.

**Questions:**

1. See weaknesses.
2. The algorithm does not show significant improvement on the Airport dataset. It would be useful to provide a more detailed analysis of the reasons behind this.
3. The experimental setup for rotation in Table 1 lacks sufficient explanation. It would be helpful to clarify what specific settings or transformations were applied during this experiment, as well as its relevance to the overall analysis.
4. While the experiments demonstrate the effectiveness of the method, the analysis is not comprehensive enough. The paper could be improved by including additional robustness or sensitivity experiments to assess the method’s performance under varying conditions.

---

> ### Author Response · Authors · 2025-11-21
>
> **W1.** Limited Novelty in Methodology. While the proposed method is a valuable contribution, the use of adversarial alignment and feature reweighting is not entirely new. These techniques have been explored in several other papers on DA and GDA. The paper could provide a more detailed comparison to recent methods such as other MoE-based methods to better position DACS within the current landscape of domain adaptation methods.
>
> **AW1.** We thank the reviewer for the feedback. The novelty of DACS lies not in the choice of alignment tools, but in a new *problem formulation*, a *new decomposition* of covariate shift (CS), and a *theory-driven dual-alignment paradigm* that has not been explored in prior GDA research.
>
> **(1) Conceptual Novelty: A New and Principled Decomposition of CS in GDA**
>
> Although adversarial alignment and reweighting themselves are known techniques, **DACS introduces a new conceptual and theoretical foundation for GDA**. Specifically, we propose a **new decomposition of CS** in graph domain adaptation into two *distinct and independently quantifiable* components:
>
> * **Feature Shift (FS)**
> * **Feature-Conditional Structure Shift (FCSS)** — formally defined in Line 147
>
> The proposed FCSS is a *new shift type* that captures **how feature–structure dependency mechanisms differ across domains**.
> This factor is *clearly present in real GDA benchmarks* (Fig. 1), yet **completely overlooked by existing methods**.
>
> More importantly, this decomposition is not conceptual only—it leads to a **new target risk bound (Theorem 3.1)**, in which FS and FCSS induce *two separate error terms* (Term II and Term III).
>
> Such a decomposition does not appear in prior GDA analyses and **logically necessitates** a *dual-alignment procedure*:
>
> * **Section 4.1:** adversarial feature alignment → reduces FS-induced error
> * **Section 4.2–4.3:** layer-wise structural reweighting + representation alignment → reduces FCSS-induced error
>
> Thus, DACS is not a heuristic stacking of known tools but a **theoretically derived alignment paradigm**. It contributes:
>
> * **a novel decomposition of CS into FS and the newly introduced FCSS**,
> * **a new error-characterization framework**,
> * **and a dual-alignment algorithm that is an inevitable consequence of this theory**.
>
> The theory–algorithm coupling is the core novelty of DACS. Within this new framework, alignment tools such as adversarial networks, MMD, OT, or other metrics serve only as *instantiations*, underscoring the generality and extensibility of DACS.
>
> **(2) Algorithmic Novelty: A New Reweighting Mechanism Beyond Prior GDA Methods**
>
> While DACS uses “reweighting,” its mechanism and purpose fundamentally differ from the baseline methods StruRW and PairAlign.
>
> **(i) DACS is label-free; prior reweighting methods require target pseudo-labels**
>
> StruRW and PairAlign estimate label-conditional distributions $P(A \mid Y)$, requiring target-domain pseudo-labels, which are oftern unstable and noisy under domain shift.
>
> In contrast, **DACS estimates ($P(A | X)$) solely through feature similarity**, meaning:
>
> * no pseudo-labels are needed,
> * fully unsupervised adaptation on the target,
> * improved robustness.
>
> This captures a feature-driven structural shift unexplored in prior work.
>
> **(ii) DACS uses dynamic, layer-wise structural reweighting; prior works use static weights**
>
> Existing methods build a **single, static reweighted adjacency matrix**, shared across all layers, which fails to capture evolving latent semantics during GNN propagation.
>
> DACS addresses this by computing at every layer:
> $$
> w_{uv}^{\ell} = \frac{P_T(A_{uv} \mid H^{\ell})}{P_S(A_{uv} \mid H^{\ell})},
> $$
> allowing:
>
> * semantic adaptation as embeddings evolve,
> * more expressive message passing,
> * theoretical support from Theorem 4.1.
>
> This dynamic mechanism yields stronger performance in both synthetic and real datasets.
>
> **(3) Complementarity to MoE-Based Adaptation Methods**
>
> MoE-based DA primarily addresses **domain heterogeneity** using expert specialization, and is mostly designed for multi-source DA rather than the GDA problem with one source and one target.
>
> DACS tackles a different challenge: a **theoretically principled decomposition of CS** and alignment of FS and FCSS with provable risk reduction. Thus, the two paradigms are complementary.
>
> We will expand the comparison in related work.
>
> **Summary of Novel Contributions**
>
> DACS is novel not because of its use of adversarial alignment or reweighting, but because it introduces:
>
> * **a new decomposition of covariate shift into FS and FCSS**,
> * **a new target-risk bound separating FS- and FCSS-induced errors**,
> * **a theoretically motivated dual-alignment framework**,
> * **a label-free, feature-conditional, dynamic structural reweighting mechanism**,
> * **and a general algorithmic paradigm that subsumes existing alignment techniques**.
>
> We will revise the paper to more clearly highlight these conceptual and theoretical contributions.

---

> ### Author Response · Authors · 2025-11-21
>
> **W2.** The writing and presentation of the paper could benefit from further polishing. The logical flow is not always clear, and the use of notations is dense, which might hinder reader comprehension. Additionally, the figures and tables could be enhanced to improve clarity and better illustrate key concepts and experimental results.
>
> **A2.**
>
> We thank the reviewer for the constructive feedback. We have carefully improved the writing, strengthened the figures, clarified the experimental setup, and expanded analysis to ensure that the paper is easier to follow while more clearly conveying the core contributions of DACS.
>
> **(1) On Mathematical Clarity and Logical Flow**
>
> DACS is built upon a **new conceptual formulation of covariate shift in GDA** and a **theoretically derived dual-alignment framework**.
> As a result, the method naturally requires a more formal presentation than typical GDA work.
>
> Following the reviewer’s suggestion, we have:
>
> * reorganized all definitions in a more top-down manner,
> * added intuitive explanations before formal expressions,
> * improved notational consistency,
> * inserted short guiding paragraphs at key transitions.
>
> These changes substantially enhance readability while maintaining the rigor required for our theoretical results.
>
> **(2) Improved Figures: Clear Illustration of FS, FCSS, and Reweighting**
>
> To make the proposed concept of **Feature-Conditional Structure Shift (FCSS)** more intuitive, we added a concise illustrative example consisting of two graphs with binary features, as visualized in Figure (https://anonymous.4open.science/r/illustration-of-reweighting-process).
>
> **Left subfigure: visualizing FS and FCSS**
>
> * Nodes 1 and 3 change their features across domains, which implies **Feature Shift (FS)**.
> * We compute, for each feature distance $d(H_u, H_v)$, the empirical edge probability:
>   $$
>   P(A_{uv}=1 \mid d(H_u,H_v)=j).
>   $$
>
> Example:
>
> * In the **source** domain, 5 node pairs ($1$-$3$, $2$-$3$, $2$-$4$, $3$-$5$, $4$-$5$) have distance 1, among which 3 are edges, which yields the edge probability $3/5$.
> * In the **target** domain, all four distance-1 pairs are connected by edges, which yields the edge probability $4/4 = 1$.
>
> This simple calculation provides a **direct, visual demonstration of FCSS**, showing that even with the same features, **feature–structure dependency differs** across domains.
>
> **Right subfigure: illustrating how reweighting corrects FCSS**
>
> We compute for each pair:
> $$
> w_{uv}=\frac{P_T(A_{uv}=1 \mid d(H_u,H_v))}{P_S(A_{uv}=1 \mid d(H_u,H_v))},
> $$
> and visualize the resulting **upweighted** (thick) and **downweighted** (thin) edges.
>
>
> This clearly shows *how* and *why* DACS performs structural alignment—
> DACS reinforces edges that are more likely in the target graph and attenuates those that are less likely—thus directly correcting FCSS.
>
> Together, these new visualizations make the core concepts of DACS transparent and easy to understand.
>
> **(3) Clarifying the Synthetic Experiment Design**
>
> To rigorously evaluate how GDA methods behave under different shift types,
> we designed the synthetic datasets to *explicitly and independently* control:
>
> * **Only Feature Shift (FS)**
> * **Only Structure Shift (SS)**
> * **Their combination**
>
> **FS manipulation (Row 1 of Table 1)**
>
> Target features are rotated by:
>
> * $0^\circ$: no FS
> * $30^\circ$: mild FS
> * $60^\circ$: severe FS
>
> **SS manipulation (Row 2 of Table 1)**
>
> Target intra-class/inter-class edge probabilities $(p,q)$ are set to:
>
> * identical to source $(0.02,0.002)$: **no SS**
> * $(0.02,0.010)$: **mild SS**
> * $(0.015,0.016)$: **severe SS**
>
> This orthogonal construction allows us to isolate each component of CS and examine their interaction—something rarely done in prior GDA work. The specific procedures used to implement FS, SS, and their combination are further detailed in our answer A3 to question Q3.
>
> **Key findings validating our decomposition**
>
> * Under **FS-only** (column 3), baseline methods degrade only slightly: our improvement over the second-best method GAA that is specialized design to address FS is **4.24\%**. Once **SS is added** (columns 8–9), the baseline methods including GAA collapse dramatically, while our improvement rises to **8.41\%** and **12.40\%**.
>
> * Under **SS-only** (column 5), baseline methods degrade only slightly: our improvement over the second-best method PairAlign that is specialized design to address SS is **1.91\%**. Once **FS is added** (columns 7 and 9), the baseline methods including PairAlign collapse dramatically, while our improvement rises to **3,81\%** and **12.40\%**.
>
> This strongly supports our core claim:
>
> * **Validate the effectiveness of the new decomposition of CS into FS and FCSS.**
> * **A method specifically designed to address only FS or only SS cannot remain robust under mixed shifts.**

---

> ### Author Response · Authors · 2025-11-21
>
> **(4) Strengthened Analysis on Real-World Benchmarks**
>
> **DBLP/ACM and ArXiv (Tables 2–3)**
>
> DACS consistently outperforms classical adversarial alignment approaches (e.g., AdaGCN) by **more than 5\% on average**.
>
> Compared with the second-best method, DACS improves accuracy by **over 2\%** on most tasks.
> Especially for **A-D**, the improvement reaches **7.84\%**, indicating that **FCSS is a major source of error** in citation networks and existing methods fail to mitigate it effectively.
>
> **Airport dataset**
>
> Because node features in the Airport dataset are simple one-hot encodings of node degrees, they contain no semantic information beyond the degree itself. Although degree distributions differ across domains, the feature-conditional mechanism governing edge formation is effectively fixed: once the degree of each node is given, the number of edges it forms is deterministic and does not vary across domains. As a result, FCSS is extremely weak or nearly absent.
>
> Since our method is explicitly designed to correct both feature shift (FS) and feature-conditional structure shift (FCSS)—whereas baselines do not explicitly model or correct FCSS—the extremely weak FCSS in the Airport dataset causes all methods to behave much more similarly than on other datasets. This matches our theoretical prediction: the advantages of DACS arise from correcting both FS and FCSS, and when FCSS is negligible, all approaches—including DACS—are expected to exhibit comparable performance.
>
> **Summary**
>
> We have significantly improved the clarity of the writing, strengthened visual explanations, clarified experimental setup, and expanded real-world analyses.
> The empirical results across synthetic and real datasets coherently support our main theoretical contribution:
>
> * FCSS is an essential and previously overlooked component of covariate shift in graphs;
> * dual alignment is necessary to jointly correct FS and FCSS;
> * and DACS is the first method to provide a principled solution to both.
>
> **Q1.** See weakness.
>
> **A1.** See AW1 & AW2.
>
> **Q2.** The algorithm does not show significant improvement on the Airport dataset. It would be useful to provide a more detailed analysis of the reasons behind this.
>
> **A2.** We appreciate the reviewer’s observation. The relatively small improvement on the **Airport** dataset is in fact consistent with our theoretical formulation.
>
> The limited improvement on the Airport dataset primarily stems from the special nature of its node features. In this dataset, each node is represented by a one-hot vector indicating its degree. The feature dimension equals the maximum node degree, and each feature vector contains a single non-zero entry at the position corresponding to that node’s degree. As a result, the node feature matrix $X$ is essentially a direct encoding of the graph structure $A$.
>
> In the Airport dataset, feature shift exists because the distribution of node degrees is different between domains. However, since features only encode the information about the node degree, it contains no semantic or latent information beyond the degree itself. Since degree is a purely structural property,
> once $X$ is fixed, many structural properties are already fully determined.
> Specifically, given the degree-based feature matrix $X$, the number of edges adjacent to each node is predetermined and identical across domains. In other words, the way features influence the structure is already fixed by the degree encoding, leaving little room for domain-specific variation.
>
> Our method is specifically designed to correct FS and FCSS, two types of domain shift that arise when node attributes change across domains and when structural patterns depend on features differently across domains. Since one of the two major components of CS is weak in this dataset, the FCSS-mitigation module of DACS is **not expected** to produce substantial gains. Therefore, the Airport dataset exhibits naturally smaller performance gaps between methods, which aligns precisely with our theoretical prediction.
>
> In summary, the small improvement is not a limitation of DACS, but a **direct consequence of the dataset’s characteristics**, where FCSS is weak, and thus cannot be further reduced.

---

> ### Author Response · Authors · 2025-11-21
>
> **Q3.** The experimental setup for rotation in Table 1 lacks sufficient explanation. It would be helpful to clarify what specific settings or transformations were applied during this experiment, as well as its relevance to the overall analysis.
>
>
> **A3.**
>
> Thank you for pointing this out. We clarify the rotation-based experimental setup in Table 1 and its relevance to our analysis.
>
> **(1) Constructing Feature Shift (FS) via Controlled Rotation**
>
> We first generate *source-domain node features* from a Gaussian mixture model (GMM).
> To introduce **feature shift (FS)** in the target domain, we:
>
> 1. sample node features independently from the same GMM for both domains,
>
> 2. then apply a $2 \times 2$ rotation matrix (with rotation angles $30^\circ$ or $60^\circ$) to the sampled target-domain node features, while keeping the source-domain features unrotated.
>
> 3. in both domains, we cluster features $X$ into three groups using $k$-means algorithm to assign node labels $Y$,
>
> This rotation preserves pairwise distances but **changes the global feature direction**, producing a *controlled and interpretable* degree of FS. Since the source features are unrotated:
>
> * **$60^\circ$** rotation → **severe FS**,
> * **$30^\circ$** rotation → **mild FS**,
> * **no rotation** → **no FS**.
>
> This directly corresponds to the first row of Table 1.
>
> **(2) Constructing Feature-Conditional Structure Shift (FCSS)**
>
> To model structural dependencies conditioned on features, we:
>
> 1. sample nodes features independently from the GMM for both domains,
>
> 2. cluster features $X$ into three groups using $k$-means algorithm to assign node labels $Y$ for both domains,
>
> 3. generate graphs using a stochastic block model (SBM) with intra-/inter-class probabilities $(p, q)$. We fix the source domain at $(p, q) = (0.02, 0.002)$.
> For the target domain, we vary:
>
> * **$(0.02, 0.010)$** → **mild FCSS**,
> * **$(0.015, 0.016)$** → **severe FCSS**,
> * **$(0.02, 0.002)$** → **no FCSS**.
>
> These settings compose the second row of Table 1. In SBM, the variation in $(p,q)$ across domains reflects differences in the mechanism by which labels influence the structure. Moreover, since labels in both domains are generated by clustering domain-invariant features, the variation in $(p,q)$ also captures differences in the mechanism by which features influence the structure, i.e., FCSS. Therefore, more discrepancy between $(p,q)$ yields more severe FCSS.
>
>
> **(3) Constructing both FS and FCSS:**
>
> To construct a case where both FS and FCSS are present, we only need to add one additional step to the existing procedure for generating FCSS. Specifically, between its step 1 and 2, we insert a feature-rotation operation on the target features drawn from the GMM. This rotation step corresponds exactly to the step 2 in the procedure used to generate FS. Thus, by augmenting the FCSS construction pipeline with this intermediate rotation step, we simultaneously introduce both FS and FCSS.
>
>
> **(4) Why This Design Matters**
>
> By **orthogonally combining**:
>
> * three FS levels ($0^\circ$, $30^\circ$, $60^\circ$), and
> * three FCSS levels (none, mild, severe),
>
> we obtain nine synthetic datasets where FS and FCSS can be **precisely controlled and varied independently**.
> This enables us to systematically evaluate DACS and baselines under:
>
> * FS only (columns 2-3),
> * FCSS only (columns 4-5),
> * both FS and FCSS (the most challenging scenario, columns 6-9).
>
> This controlled setup directly supports the theoretical decomposition of CS into FS and FCSS in our paper and allows us to isolate where each method succeeds or fails under different cases. Additional result interpretation is provided in Answer AW2.

---

> ### Author Response · Authors · 2025-11-21
>
> **Q4.** While the experiments demonstrate the effectiveness of the method, the analysis is not comprehensive enough. The paper could be improved by including additional robustness or sensitivity experiments to assess the method’s performance under varying conditions.
>
>
> **A4.**
>
> We thank the reviewer for raising this point. To provide a more comprehensive evaluation, we conducted **additional robustness and sensitivity experiments** targeting noisy FS, noisy FCSS, and reduced feature separability. These experiments are directly aligned with our theoretical framework, which identifies FS and FCSS as the two principal sources of error.
>
> **(1) Robustness to Noisy Feature Shift (FS)**
>
> To simulate noisy FS, we perturb each feature dimension by adding i.i.d. Gaussian noise, where the noise standard deviation is set to $0.1$, $0.3$, or $0.5$ times the original standard deviation. This creates increasingly severe distortions in the feature space, weakening the alignment signal for all methods.
>
> **Results (A→D task):**
>
> | Method   | 10\% Noise      | 30\% Noise      | 50\% Noise      |
> | -------- | -------------- | -------------- | -------------- |
> | StruRW   | 70.79±10.24    | 69.41±5.02     | 70.59±5.65     |
> | **DACS** | **84.17±2.20** | **82.14±5.93** | **81.83±4.52** |
>
> Despite feature corruption, DACS maintains a large improvement margin over StruRW, confirming that our feature-alignment module remains stable even under noisy FS.
>
> **(2) Robustness to Noisy FCSS**
>
> To evaluate robustness against structural noise, we modify the adjacency matrix by randomly removing a fixed percentage (10\%, 20\%, 30\%) of existing edges and adding an equal number of random edges. This directly perturbs the structure-conditioned distribution $P(A \mid X)$, simulating noisy FCSS.
>
> **Results (A→D task):**
>
> | Method   | 10\% Noise   | 30\% Noise    | 50\% Noise    |
> | -------- | -------------- | -------------- | -------------- |
> | StruRW   | 67.88±3.80     | 63.80±6.50     | 60.17±7.15     |
> | **DACS** | **72.69±4.39** | **66.76±4.42** | **60.99±2.62** |
>
> DACS consistently outperforms the strong baseline StruRW across all noise levels, demonstrating that our **layer-wise reweighting** remains effective even when structural information is corrupted.
>
> **(3) Sensitivity to Reduced Feature Separability**
>
> To further assess performance under more challenging conditions, we constructed an alternative synthetic dataset using three Gaussian components with **closer means and smaller variances**:
> $$
> P_0 = \mathcal{N}([-0.8, 0], 0.64 I),
> \quad
> P_1 = \mathcal{N}([0.8, 0], 0.64 I),
> \quad
> P_2 = \mathcal{N}([0, 0.8], 0.64 I),
> $$
> which produces **less separable clusters** compared to Table 1. This reduces the distinguishability of both features and structures, making both FS and FCSS harder to identify.
>
> **Results:**
>
> | Rotate / Edge Prob | $30^\circ$ / No SS | $60^\circ$ / No SS | $0^\circ$ / $(0.02,0.01)$ | $0^\circ$ / $(0.015,0.016)$ | $30^\circ$ / $(0.02,0.01)$ | $30^\circ$ / $(0.015,0.016)$ | $60^\circ$ / $(0.02,0.01)$ | $60^\circ$ / $(0.015,0.016)$ |
> | ------------------ | --------- | --------- | ---------------- | ------------------ | ----------------- | ------------------- | ----------------- | ------------------- |
> | StruRW             | 99.08     | 67.39     | 77.23            | 66.65              | 59.61             | 47.15               | 50.79             | 39.64               |
> | PairAlign          | 96.32     | 63.00     | 92.24            | 79.59              | 72.93             | 66.56               | 55.23             | 47.62               |
> | **DACS**           | **99.98** | **85.23** | **95.09**        | **87.89**          | **83.78**         | **77.72**           | **65.91**         | **52.55**           |
>
> Across all configurations—especially those combining severe FS and severe FCSS—DACS consistently retains a significant advantage.
>
> **(4) Conclusion**
>
> Across noisy FS, noisy FCSS, and reduced separability scenarios, DACS consistently outperforms strong baselines. These results demonstrate that our **dual alignment framework is inherently robust**, even when domain shifts are imperfect, distorted, or difficult to detect—validating the central motivation behind decomposing covariate shift into FS and FCSS.

---

> > ### Comment · Reviewer_8DxB · 2025-11-21
> > **I will raise my score.**
> >
> > I appreciate the authors’ careful and thorough responses to all of my questions. In particular, the reply to Q4 was commendable—the authors provided extensive additional experiments, and their restatement of the key points further clarified the contributions of this work. Although I have not seen the final revised version of the paper, I trust that if the authors have polished the writing and incorporated the relevant elements from the rebuttal, this would amount to a solid piece of research.
> >
> > After careful consideration, I intend to raise my overall score, including the ratings for soundness and contribution. I hope the authors will further refine and elaborate on the parts of the paper that may cause confusion.

---

> > > ### Author Response · Authors · 2025-11-27
> > > **Response to Reviewer’s Follow-Up Assessment**
> > >
> > > We sincerely thank the reviewer for the thoughtful and positive follow-up assessment. We are especially grateful for the recognition of our response to Q4 and the acknowledgment of the value added by the additional experiments and clarifications in the rebuttal. We also appreciate the reviewer’s willingness to raise the overall score as well as the ratings for soundness and contribution.
> > >
> > > We have uploaded a revised manuscript and will continue to refine the writing and integrate relevant elements from the rebuttal into the final version to further enhance clarity and avoid potential sources of confusion. We are grateful for the reviewer’s constructive feedback, which has substantially strengthened the quality and rigor of the paper.

---

### Author Response · Authors · 2025-11-30
**Rebuttal Summary**

Dear Program Chairs, Senior Area Chairs, Area Chairs, and Reviewers,

We sincerely appreciate the tremendous efforts of the Program Chairs, Senior Area Chairs, and especially the Area Chairs in coordinating the review process. We also thank all reviewers for their constructive evaluations.

We believe we have thoroughly addressed all concerns raised during the review. Importantly, **Reviewer 8DxB** and **Reviewer Ng5Z** **increased their scores** (**4 → 6** and **6 → 8**, respectively) *before* the security incident, each providing clear justification. Therefore, the score **prior to the incident should be 8–6–6–2**.

We kindly hope that the new AC can consider these raised scores. Due to the unexpected interruption, Reviewers **B1Yv** and **mmvb** did not have an opportunity to comment further, which was unfortunate as our clarifications might have further improved the evaluations.

Additionally, we note that the review from Reviewer mmvb (score 2) contains factual inaccuracies. For example, (i) the claim that our experiments are "insufficient" is not accompanied by specific evidence; (ii) the statement that "datasets in the paper are not commonly used in GDA" contradicts recent ICML/NeurIPS/CIKM papers; and (iii) the reviewer overlooked the parameter analysis in Appendix C.3, explicitly cited in Section Experiment of our initial submission. These issues suggest this review should be weighted with caution.

We summarise the key points from each reviewer below.

**Reviewer 8DxB (4 → 6).**
The reviewer explicitly acknowledged our “careful and thorough responses” and wrote that "in particular, the reply to Q4 was commendable". Accordingly, the reviewer raised the overall score from 4 to 6 with improved ratings for soundness and contribution on the 21st, the day we submitted our full rebuttal.

**Reviewer Ng5Z (6 → 8).**
The reviewer raised one remaining concern about W1 on the 26th at 12:37am EST. We clarified on the 27th at 1:13am that the reviewer’s proposed scenario falls outside the scope of GDA. The reviewer realised the difference between GDA and domain heterogeneity and wrote that “the proposed reweighting strategy plays a meaningful and impactful role within the scope of GDA”, raising the score from 6 to 8 on 27th at 4:52am—before the security incident was reported at 10:09am.

**Reviewer B1Yv (score 6, no further discussion).**
* For W1, we clarified that our complexity is approximately $O(Nk)$, not $O(N^2)$, and we provided additional scalability experiments, which also resolved the same issue raised by Reviewer Ng5Z.
* For W2, we justified the approximation of conditional edge probability.
* For W3, we demonstrated hyperparameters' robustness with further explanations and experiments.
We refer the AC to our rebuttal for full details.

**Reviewer mmvb (score 2, no further discussion).**
For W1, the reviewer mmvb stated that our experiments were insufficient but did not specify which aspects were lacking. We clarified that our experimental design is systematic, multi-layered, and directly aligned with validating both our theoretical decomposition and our dual-alignment framework. Moreover, all other reviewers, 8DxB, Ng5Z, and B1YV, acknowledged the validity and strength of our experiments. Specifically:

* Reviewer 8DxB wrote "This empirical evidence supports the claims made by the authors and demonstrates that DACS can be applied successfully across different domains in GDA".

* Reviewer Ng5z wrote "Results show that DACS improves robustness when feature and structural shifts co-exist, supporting the paper’s central claim empirically. The ablation study is informative".

* Reviewer B1YV wrote "The experimental results are comprehensive and convincing. DACS shows state-of-the-art performance across all datasets. The synthetic experiments (Table 1) are particularly strong, ..." and "The ablation analysis (Table 5) clearly demonstrates that both the FS and FCSS components are crucial and complementary."

For W2, we clarified that the datasets we used are **standard benchmarks** widely adopted in recent ICML/NeurIPS/CIKM papers, and we additionally conducted experiments on the datasets requested by the reviewer. We also clarified that **the airport dataset does contain node features** and has been used in prior work studying feature shift.
For W3, we added the requested ablation for the conditional representation alignment module.
For W4, it appears the reviewer overlooked our parameter analysis in Appendix C.3, which was already cited in the main text of the initial submission. We also reiterated this analysis clearly in our rebuttal.

In the revised version, we improved the writing, strengthened the figures, and incorporated expanded empirical results, complexity analysis, and a brief discussion of the relationship between GDA and multi-domain graph generalization under domain heterogeneity.

Best regards,
Authors

---

### Meta-Review · Area_Chair_zY1s · 2026-01-06

**Summary:**

This paper introduces DACS, a method aimed at improving GDA by addressing covariate shift. DACS aligns both Feature Shift (FS) and Feature-Conditional Structure Shift (FCSS) using adversarial feature alignment, adaptive reweighting, and final-layer representation alignment. The authors report that DACS outperforms existing methods through extensive experiments across multiple datasets. However, reviewers have identified several limitations in the current version, including limited methodological novelty, reliance of the reweighting estimation on strong approximations, and a relatively narrow experimental evaluation. Thus, I recommend rejecting this paper.

**Reviewer Concerns:**

The rebuttal addresses concerns regarding performance validation and the explanation of DACS’s alignment mechanisms, providing additional experiments and clarifications. However, several issues remain outstanding, including limited methodological novelty, reliance on strong approximations in the reweighting estimation, and a still relatively narrow scope of experimental evaluation.

**Reviewer Scores:**

The rebuttal offers limited clarification on the methodology but does not alleviate concerns about novelty. The reviewer’s score might likely remain unchanged.

---

### Decision · Program_Chairs · 2026-01-26

Reject